# Basal association of a transcription factor favors early gene expression

**Sandrine Pinheiro[1], Mariona Nadal-Ribelles[2,3], Carme Solé[2,3], Vincent Vincenzetti[1], Yves Dusserre[1], Francesc Posas[2,3], Serge Pelet[1]***

**1** Department of Fundamental Microbiology, University of Lausanne, Lausanne, Switzerland,
**2** Department of Medicine and Life Sciences, Universitat Pompeu Fabra, Barcelona, Spain, **3** Institute for Research in Biomedicine (IRB Barcelona), The Barcelona Institute of Science and Technology, Barcelona, Spain

\* serge.pelet@unil.ch

## Abstract

Responses to extracellular signals via Mitogen-Activated Protein Kinase (MAPK) pathways control complex transcriptional programs where hundreds of genes are induced at a desired level with a specific timing. Gene expression regulation is largely encoded in the promoter of the gene, which harbors numerous transcription factor binding sites. In the mating MAPK pathway of *Saccharomyces cerevisiae*, one major transcription factor, Ste12, controls the chronology of gene expression necessary for the fusion of two haploid cells. Because endogenous promoters encode a large diversity of Ste12 binding sites (PRE), we engineered synthetic promoters to decipher the rules that dictate mating gene induction. Conformations of PRE dimers that allow efficient gene expression were identified. The strength of binding of Ste12 to the PRE and the distance of the binding sites to the core promoter modulate the level of induction. The speed of activation is ensured by favoring a basal association of Ste12 by using a strong dimer of PRE located in a nucleosome depleted region.

## Author summary

During development, cell fate decisions allow pluripotent cells to differentiate into various cell types. This process requires cells to integrate signals from their surroundings to initiate a complex transcriptional program. Budding yeasts can also undergo cell fate decisions. In presence of mating pheromones, haploid yeasts can activate a signaling pathway which can ultimately lead to the fusion of two haploid cells to form a diploid.

One transcription factor, Ste12, controls this mating transcriptional program. The promoters of these 200 upregulated genes display a large diversity in the organization of Ste12 binding sites. Therefore, it is challenging to decipher how Ste12

**Data availability statement:** Analyzed single cell measurements are available on Zenodo (DOI: 10.5281/zenodo.14438715). The raw images of this study are available from the Data Stewardship Biomed Unit (DSBU) of the Faculty of Biology and Medicine of the University of Lausanne (https://wp.unil.ch/dsbu/) E-mail contact to the DBSU is dsbu@unil.ch. Code Availability The code is available on GitHub (https://github.com/sergepelet/YeastQuantX_V2). Specific analysis scripts are provided on the Zenodo repository along with the data.

**Funding:** Work in the Pelet lab is funded by the Swiss National Science Foundation (SNSF, 31003A_182431 and 320030-231544) and the University of Lausanne. This work in FP's lab was founded by PID2021-124723NB-C21 from MICIU/AEI /10.13039/501100011033 and ERDF/EU and from the Ministry of Science, Innovation and Universities through the Centres of Excellence Severo Ochoa Award. FP is a recipient of an ICREA Acadèmia award (Government of Catalonia, 2024 ICREA 00084). The Ramon y Cajal Program (Spanish Ministry of Science, RYC2021-033520-I) awarded to MNR. The funders had no role in study design, data collection and analysis, decision to publish, or preparation of the manuscript.

**Competing interests:** The authors have declared that no competing interests exist

regulates the level and the timing of gene expression. To simplify this problem, we have generated synthetic promoters, where the configuration of Ste12 binding sites on the DNA can be controlled. We have identified which conformations of binding site dimers allow a functional association of the transcription factor. In addition, we have also shown that the basal association of Ste12 to the promoter is important for the fast gene induction. An unfavorable configuration of Ste12 binding sites or the presence of nucleosomes restrict the access of the transcription factor to the DNA and results in a slower expression.

## Introduction

*De novo* protein synthesis plays a central role in all cellular functions. This process can be controlled by internal regulatory inputs emanating from the cell cycle machinery [1,2], from fluctuations in circadian rhythms [3] or from oscillations in the metabolic state [4]. In addition, extracellular cues such as stresses, nutrients or hormones can stimulate gene expression. In all eukaryotic cells, Mitogen-Activated Protein Kinase (MAPK) pathways play an essential role in transducing extracellular information into a cellular response, which generally includes the production of new proteins [5,6]. The gene induction process may be transient, in order to adapt to a new stressful environment [7]. However, if the protein production is sustained, it can profoundly modify the cellular physiology by altering its entire proteome [8]. Cell fate decision systems implicated in cellular differentiation mechanisms rely on this *de novo* protein production to transform naive pluripotent cells into differentiated cells that will ultimately give rise to the different parts of a multicellular organism.

Despite their relative simplicity, unicellular organisms can also take complex decisions. The budding yeast *Saccharomyces cerevisiae* induces diverse transcriptional programs via MAPK cascades that allow this microorganism to make appropriate decisions in response to a specific stimulus. As an illustration, in low nutrient conditions, both haploid and diploid cells can alter their growth pattern to form pseudohyphae [9]. Under more drastic nutrient limitations, diploid cells begin to sporulate [10]. In rich medium and in the presence of a mating partner, haploid cells can commit to mating to produce diploid cells [11,12].

During the mating process, cells of opposing mating types (MATa or MATα) communicate by secreting pheromones (a- or α-factor, respectively). At the cell surface, binding of pheromones stimulates a G-protein coupled receptor which in turn activates the three-tiered kinase cascade [11,12] (S1 Fig). The MAPKs, Fus3 and Kss1, release the inhibition of Dig1 and Dig2 on the transcription factor (TF) Ste12 [13]. This step is critical for activating the mating transcriptional program, resulting in the up-regulation of more than 200 genes [14]. Note that a small fraction of these genes, which are typically cell-type specific, depend on co-activators such as Mcm1, a1 or α1 and α2 [15,16]. The mating transcriptional response promotes the arrest of the cell cycle and the formation of the mating projection, two processes which are necessary to ensure a robust mating of the partner cells.

The proteins involved in the various stages of the mating process are tightly regulated in their expression levels and dynamics by Ste12. Genes involved in the early phase of mating, such as the establishment of a pheromone gradient (*BAR1*), MAPK signal transduction (*FUS3*, *STE12*), cell cycle arrest (*FAR1*) and cell agglutination (*AGA1*) are expressed rapidly after the detection of the pheromone [17]. In contrast, genes involved in later stages, such as karyogamy (*KAR3*) and membrane fusion (*FIG1*) will be expressed with a delay [17,18]. The mechanisms which allow a single transcription factor (Ste12) to orchestrate this chronology of gene expression remain poorly understood.

The promoter sequence upstream of the protein-coding sequence regulates the level and dynamics of transcription. The promoter combines two distinct segments: the core promoter and the regulatory region [19,20]. With a typical length of 100–200 bp, the core promoter contains the TATA box which is recognized by the TATA-binding protein. This protein recruits other general transcription factors and contributes to the formation of the Pre-Initiation Complex (PIC). In yeast, the regulatory region is typically smaller than 1kb and carries Upstream Activation Sequences (UAS) recognized by TFs. Transcriptional activation is induced via the Mediator complex, which bridges the TFs and the RNA polymerase II, thereby assembling the PIC to initiate transcription [21,22].

Studies have shown that induction of mating genes requires the formation of a Ste12 homodimer on the UAS of mating promoters [23,24]. The region is recognized by the protein via the minimal DNA consensus motif TGAAAC, commonly referred to as the Pheromone Response Element (PRE) [25–27]. In addition to these consensus PREs, Ste12 also binds with lower affinity to non-consensus sites, known as PRE-like sites. In general, multiple PREs or/and PRE-like sites can be identified on the promoters of mating genes [17,28] and the arrangement of these PRE motifs controls the expression profile of a gene [17,27]. Unfortunately, defining simple rules that would allow to predict the expression pattern of mating genes is challenging because of the wide diversity in configurations present on endogenous promoters. In addition, the identification of all possible Ste12 binding sites is difficult, since it is unclear how much a PRE-like site can deviate from the consensus while retaining affinity for Ste12.

Obviously, a complex promoter architecture is not restricted to mating genes. Essentially, all endogenous promoters harbor an intricate arrangement of TF binding sites, often combining multiple TF inputs [29]. General rules determining gene expression patterns have been obtained by analyzing libraries of synthetic promoters tested for a wide diversity of binding site organization [30,31]. In general, the number of binding sites, their affinity and their distance from the core promoter can all influence the expression output. A prolonged residence time of a TF on a promoter will increase the expression output, as the chance of initiating transcription via the recruitment of the Mediator complex and formation of the PIC will rise [32]. However, it is not known whether a high transcriptional output is necessarily correlated with fast gene expression or whether these two parameters (i.e., strength and speed of induction) can be decoupled.

To decipher how the promoter sequence modulates gene expression dynamics, we have engineered synthetic promoters under the control of the TF Ste12. These promoters combine different PRE conformations to understand the parameters that regulate the transcriptional program for cell fate decisions during mating. The dynamics and level of protein production were quantified to assess how the orientation and the spacing between PRE pairs, their affinity and their location along the promoter influence the transcriptional response. Mutating the PRE consensus to lower the affinity of Ste12 for the promoter or changing the location of the sites on the promoter influenced predominantly the level of induction. However, our data show that the ability of Ste12 to associate to a promoter prior to the stimulus either by controlling by the Ste12 binding site conformation or the access to the DNA in nucleosome depleted regions favors faster gene induction.

## Results

To measure the dynamics of gene expression in the mating pathway, we use a dynamic Protein Synthesis Translocation Reporter (dPSTR) [33]. It consists of two transcriptional units: the first one encodes a fluorescent protein fused to a synthetic "zipper" coiled-coil helix (SynZip) [34] and is expressed constitutively. The second unit encodes two NLSs linked to a complementary SynZip and is placed under the control of a promoter of interest. Because SynZips form strong

heterodimers, upon the expression of the NLS, the fluorescent protein relocates into the nucleus (Figs 1A, 1B, S2A and S2B). This sensing strategy allows a rapid quantification of protein production, which can otherwise be impaired by the slow maturation kinetics of fluorescent proteins.

In our assays we combine two dPSTRs. In the yellow channel, we have a reference p*AGA1*-dPSTR$^Y$. *AGA1* encodes an agglutinin that promotes cellular adhesion of mating pairs. This gene has been shown to belong to the early gene category and is expressed at high levels, similarly to the well-established p*FUS1* reporter [17,26,35,36]. In the red channel, we monitor the dPSTR$^R$ controlled by a promoter of interest (Fig 1A and 1B). The expression outputs of a dPSTR$^Y$ and a dPSTR$^R$ controlled by the same p*AGA1* promoter are tightly correlated (S2C Fig). While p*AGA1* is activated in the entire population, other promoters are activated in a fraction of the population (S2D, S2E and S2F Fig). Importantly, the comparison of the nuclear enrichment dynamics between the reference dPSTR$^Y$ and the test dPSTR$^R$ provides an accurate measurement of the kinetics of activation of our promoter in each cell. Notably, it overcomes cell to cell fluctuations due to the cell-cycle regulated activation of the mating (S2G and S2H Fig).

## Construction of a pheromone−inducible synthetic promoter

Starting from the *AGA1* promoter, we first selected an alternative core promoter based on the *CYC1* promoter (S3 Fig). Although the *CYC1* core ensures a sizable inducibility, the expression dynamics are delayed by 5 minutes relative to p*AGA1*. This delay reinforces the idea that the core promoter identity contributes to define both the level and kinetics of gene expression [17]. In all our synthetic promoters, we will use the same p*CYC1* core promoter allowing to focus our study on the role of the regulatory sequence in the dynamics of gene expression.

The promoter activity is controlled by the regulatory region, which can contain multiple TF binding sites. In p*AGA1*, we have identified three consensus PREs and multiple PRE-like (S3A Fig). We had previously determined that two PREs spaced by 29 bp were important for the fast and high induction of the promoter upon α-factor treatment [17]. To test if these two PRE sites spaced by 29 bp are sufficient to control gene induction, we placed them in a completely synthetic context. To do so, we have selected a *CYC1* promoter, which was modified to decrease nucleosome binding [37] and where identified TF binding sites were mutated [29], as well as sequences that resembled potential PRE sites. Two PREs spaced by 29 bp placed in this synthetic context are not sufficient to induce expression (S3B and S3C Fig). The inducibility of the construct is recovered when the original sequence between the two PREs from p*AGA1* is included. This difference was attributed to the presence of a PRE-like site located 3 bp away from the second consensus PRE [24]. Engineering a *pCYC1* UAS containing two PRE-consensus sites 3 bp apart fused to the *pCYC1* core, resulted in a rapid pheromone-inducible synthetic promoter (S3B, S3C and S3D Fig). This initial synthetic construct (p*SYN*$_{3TT}$- 2PRE spaced by 3 bp in Tail-to-Tail configuration) will be used as a reference for the systematic alterations that will be performed on the PRE sites to generate our library of Ste12-dependent promoters.

## PRE site conformations

Our initial efforts to engineer a synthetic promoter have clearly demonstrated that while two PREs are required to induce gene expression, not all PRE conformations result in a functional promoter. Endogenous promoters harbor a wide diversity of PRE configurations, and it is difficult to predict which ones of these PRE or PRE-like sites contribute to the overall expression output by allowing formation of a Ste12 dimer. Dorrity *et al.* have described the canonical 3 bp tail-to-tail conformation (as found in our p*SYN*$_{3TT}$) as the most favorable binding *in vitro* for the Ste12 DNA-binding domain (DBD) fragment [24]. This PRE conformation is found on multiple endogenous promoters such as p*AGA1*, p*STE12,* p*KAR4* and p*FAR1*. However, many endogenous promoters don't harbor this element. Therefore, it is likely that other conformations can promote a functional Ste12 binding. In addition, Su *et al.* demonstrated that a head-to-tail conformation could induce gene expression, while a head-to-head positioning prevented Ste12 binding when placed in close proximity [27].

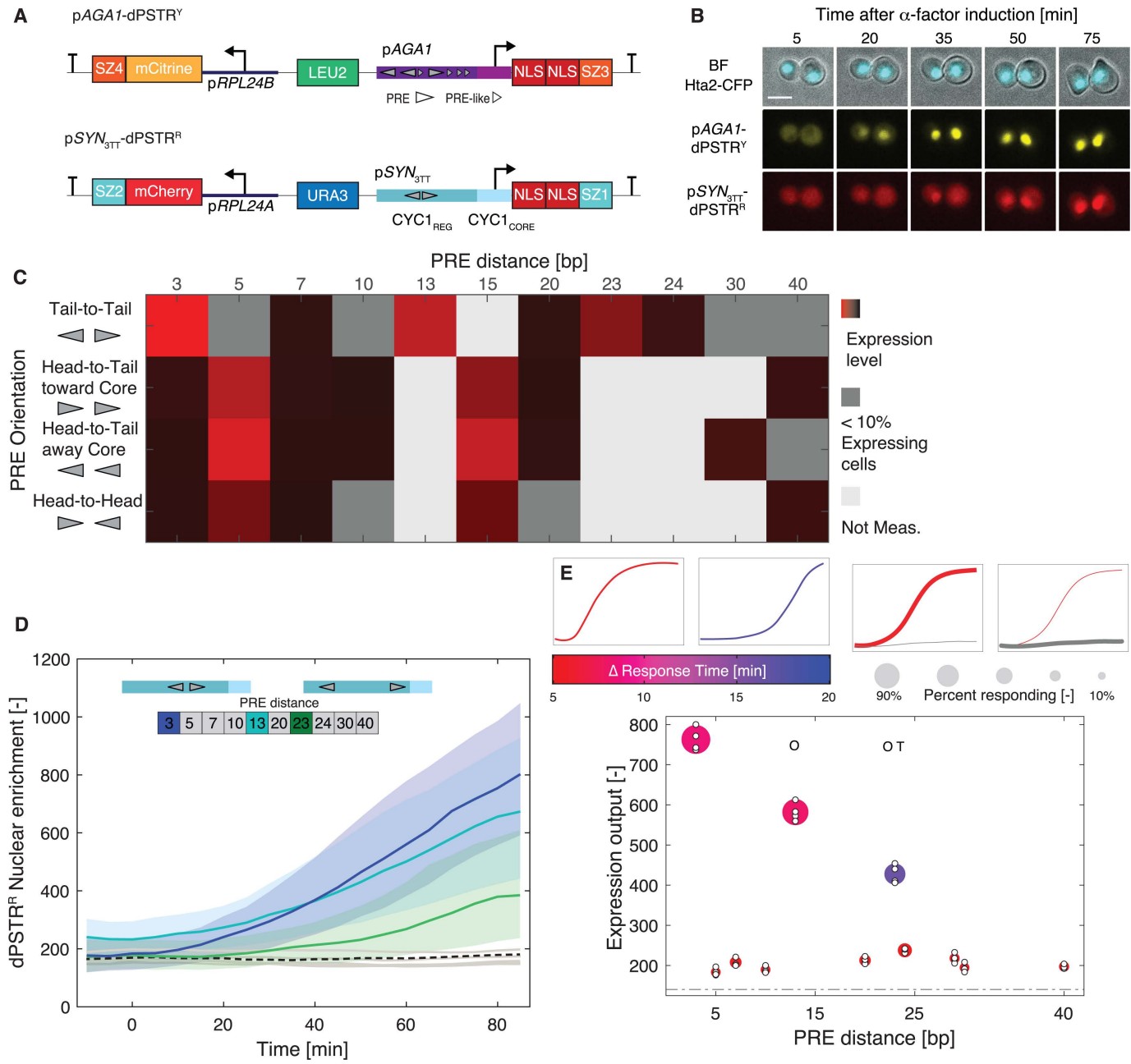

**Fig 1. Effect of the orientation and spacing of Ste12 binding sites on the expression output. A.** Scheme of the two reporters present in the strains. The reference p*AGA1*-dPSTR$^Y$ and a test construct, here the p*SYN$_{3TT}$*-dPSTR$^R$. Both constructs encode the production of a constitutively expressed fluorescent protein and an inducible peptide containing two nuclear localization sequences (NLS) and a SynZip (SZ1 or SZ3, which can form a strong heterodimer with SZ2 and SZ4, respectively). The reference dPSTR$^Y$ is under the control of the *AGA1* promoter in the yellow channel and the test dPSTR$^R$ is regulated by a synthetic promoter of interest based on a modified *CYC1* promoter. **B.** Images of cells induced with 1μM α-factor at time 0. The nuclear enrichment of the fluorescent proteins serves as a measure of promoter activity. The scale bar represents 5 μm. **C.** Matrix representing the mean expression output for p*CYC1* containing two PRE sites with four different orientations and spaced between 3 to 40 bp. The strength of the red color is proportional to the expression output of the promoter. Dark gray areas represent construct where fewer than 10% of cells overcome the expression threshold. Light gray squares are PRE conformations that were not measured. **D.** Time course of the nuclear enrichment of the dPSTR$^R$ for various distances of PRE placed in tail-to-tail orientation. The three functional conformations are plotted in blue (3 bp) light blue (13 bp) and green (23 bp). The solid lines represent the median and the shaded area, the 25- to 75-percentile of the population. Gray lines represent the median of non-functional PRE

conformations. The black dashed line is the median of the control synthetic promoter without PREs inserted. **E.** Summary graph displaying the expression output, the speed and the fraction of responding cells for various spacings of the PRE dimer placed in the tail-to-tail orientation. The color of the marker indicates the difference in response time between the synthetic promoter and the reference p*AGA1*-dPSTR$^Y$, with fast responding promoters in red and slow ones in blue as indicated by the two small schematic graphs on the left. The size of the marker represents the fraction of responding cells as depicted by the two schemes on the right. The expression output of individual replicates is indicated by small white dots. The expression threshold based on the level of p$SYN_{3TT}$ is indicated by the dashed dotted line. The letters O and T indicate a significant difference between the mean of the replicates (t-test: p-val < 0.05) in the timing of induction (T) or in the expression output (O) relative to the p$SYN_{3TT}$.

To identify functional PRE conformations, we have systematically altered the arrangement of two PRE sites in our synthetic promoter construct (Fig 1C). With PRE in tail-to-tail orientation, the promoter becomes non-functional if the distance between the PREs is extended to 5, 7 or 10 bp. Interestingly, transcriptional activity is recovered when the spacing is set at 13 bp. This distance can even be further extended to 23 bp, resulting in a low and slow expression output (Fig 1D and 1E). These functional binding conformations can be rationalized by the fact that the DNA-helix turn corresponds to 10.5 bp [38]. Therefore, spacing the two PRE by 3, 13 or 23 bp, positions the two Ste12 proteins in a similar interaction geometry. The 13 bp tail-to-tail conformation is found in the promoter of *SST2*, however, the PRE spacings of 23 was not identified in the endogenous promoters we inspected.

If the orientation of the PRE is changed to head-to-tail or head-to-head conformations, the preferred distance between PREs becomes 5 bp (Figs 1C and S4). A spacing of 15 bp is also transcriptionally active, but other spacings result in no or very low expression output. The head-to-tail conformation spaced by 5 bp is readily found in numerous promoters (p*AGA1*, p*FUS1*, p*PRM1*, p*FIG1*), while the 15 bp seems less prevalent (p*FUS2*). In contrast, the 5 bp head-to-head conformation does not seem to be frequent because we did not identify it in the two dozen endogenous promoters investigated. We also verified by deleting *STE12* that the induction of these synthetic promoters was strictly dependent on this transcription factor (S3E and S3F Fig).

Overall, these findings validate the use of synthetic promoters to identify functional binding site conformations. These results can be transferred to endogenous promoters to pinpoint the sites implicated in the mating-dependent induction. The promoter sequences can be analyzed to identify short-range interaction with specific spacings of 3 bp for tail-to-tail orientations and 5 bp for the other orientations while including a possible increment of 10 or 20 bp corresponding to one or two DNA helix turns.

## Contribution of Ste12 activation domain to gene expression

Ste12 activates transcription via specific PRE conformations. This extended set of functional PRE pairs indicates a surprising flexibility from Ste12 to homodimerize (Fig 2A). *In vitro* data have shown that the DNA binding domain (DBD) of Ste12 can dimerize to bind to two PREs [24]. However, it is difficult to imagine how the Ste12 DBD can support the various dimerization conformations identified with our synthetic promoters, which include diverse orientations and distances. Since it has also been shown that the activation domain (AD) of Ste12 can multimerize [23], we wanted to determine if this flexible region of the protein contributes to the stabilization of some of the Ste12 binding conformations that we have identified, for instance, on PREs separated by larger distances.

To test the relative contributions of the DBD and the AD of Ste12 for the inducibility of our various synthetic promoters, we replaced the region coding for the activation domain in the endogenous *STE12* locus by a fusion between the human estrogen receptor and the VP16 activation domain (EV) [39,40]. Using this construct (Fig 2B, Ste12-EV), Ste12-responsive genes will be activated by stimulating the cells with β-estradiol, which promotes the relocation of the Ste12-EV to the nucleus and by-passes the mating MAPK cascade. The strength of induction of the PRE containing promoters will be governed only by the binding of the Ste12 DBD. Indeed, no interaction from the EV domain is expected (S5A and S5B Fig). Thus, we can test if the absence of the Ste12 activation domain lowers the inducibility of our synthetic promoters.

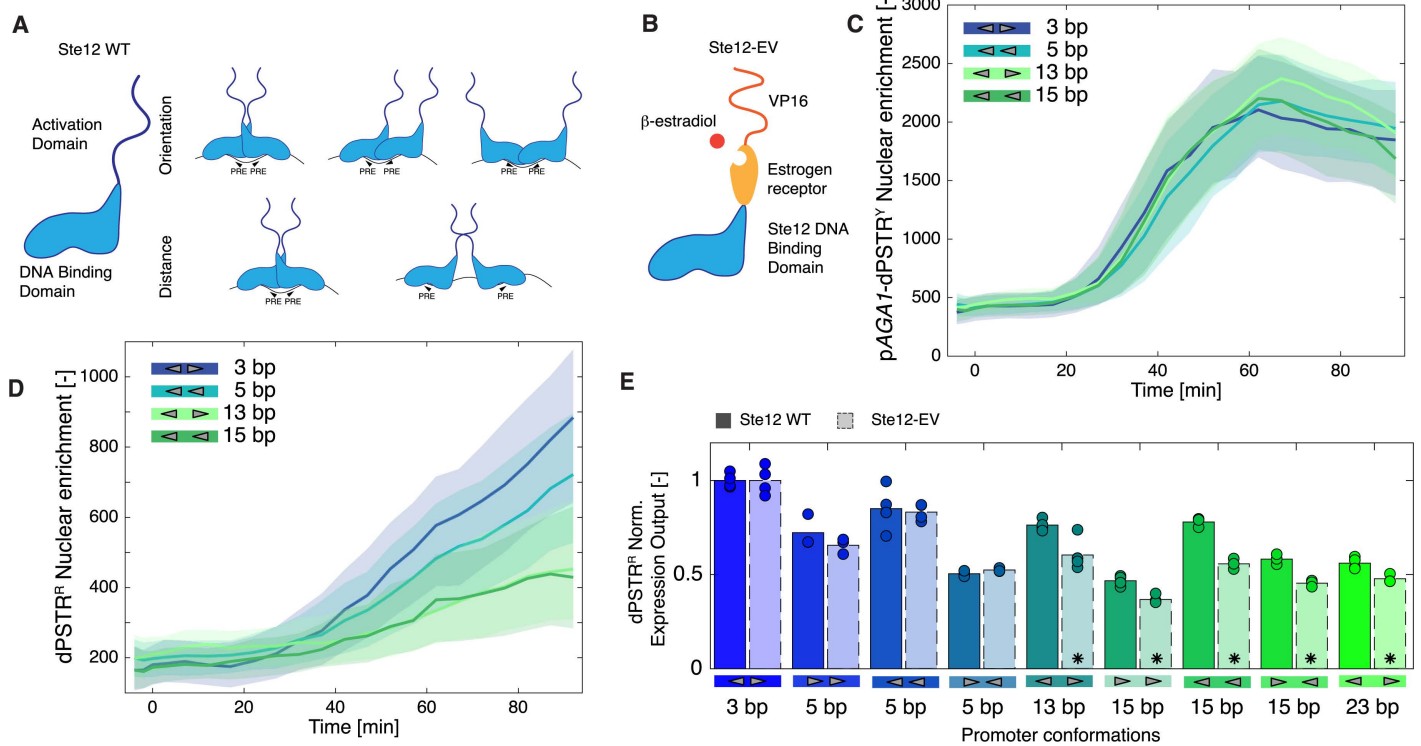

**Fig 2. Dominant role of Ste12 DNA binding domain for PRE conformation selection. A.** Schematic of the Ste12 transcription factor composed of a DNA-binding domain and a flexible activation domain. The small schemes describe the diversity of Ste12 homodimer interactions that could take place when varying the orientation or the distances of the PRE sites on the promoter. **B.** Scheme of the Ste12-EV chimeric transcription factor where the activation domain of Ste12 has been replaced by the estrogen receptor and the VP16 activation domain resulting in an β-estradiol responsive TF. **C.**, **D.** Dynamics of nuclear relocation following stimulation with 1μM β-estradiol at time 0 for strains containing the p*AGA1*-dPSTR$^Y$ (C) and various p*SYN*-dPSTR$^R$ **(D)**. The solid lines represent the median of the population and the shaded area, the 25- to 75- percentiles. **E.** Comparison of the inducibility of various PRE conformation with the Ste12 WT or the Ste12-EV (black borders). The Expression Outputs (EO) for the synthetic reporters induced by the Ste12 WT or the Ste12-EV were normalized relative to the EO of the reference p*SYN*$_{3TT}$. The bar represents the mean response of the replicates shown by the circles. A significant difference between the normalized EO of Ste12-WT and Ste12-EV based on the mean of the individual replicates is indicated by a star (t-test: p-val < 0.05).

Stimulating the cells expressing Ste12-EV with β-estradiol results in a potent, while relatively slow, activation of the p*AGA1*-dPSTR$^Y$ and of the p*SYN*$_{3TT}$-dPSTR$^R$ (Fig 2C and 2D). The time required to relocate Ste12 from the cytoplasm to the nucleus may contribute to this delay [40,41].

While the p*SYN*$_{3TT}$-dPSTR$^R$ is strongly induced in the Ste12-EV background, synthetic promoters bearing no PRE or a single PRE fail to be induced (S5C and S5D Fig). Similarly, if the PREs are spaced by 40 bp, no relocation of the dPSTR$^R$ can be observed. These control experiments demonstrate that the Ste12-EV chimeric transcription factor relies solely on the DBD domain of Ste12 for activation and if the promoter contains a single PRE or 2 PREs placed in an undesired configuration, the association of Ste12-EV to the DNA is too weak to promote transcription, despite the presence of the potent VP16 activation domain.

To determine if the activation domain of Ste12 plays a specific role in the capacity of Ste12 to promote transcription for a subset of PRE conformations, we compared the strength of induction of various synthetic promoters from wild-type Ste12 and from Ste12-EV (Fig 2E). The expression output was normalized relative to the induction of the p*SYN*$_{3TT}$-dPSTR$^R$. A significant decrease of 20–30% in the dPSTR$^R$ normalized expression output was observed for the Ste12-EV when the two PREs are spaced by 13, 15 or 23 bp. Note that the normalized p*AGA1*-dPSTR$^Y$ expression output and the fraction of

responding cells remain similar between Ste12 and Ste12-EV (S5E and S5F Fig). These results suggest that the expression output is mostly dictated by the interaction between the PRE and the DBD. However, when the two PREs are spaced by more than 5 bp, the self-interacting properties of the activation domain of Ste12 contribute to promote the expression of the downstream gene, possibly by stabilizing the formation of the Ste12 dimer when the conformation of the two PREs is not optimal.

## Affinity of the PRE sites

Suboptimal PRE-arrangements can prevent Ste12 from binding to the promoter and limit the expression output. However, even in the context of PRE-dimers with optimal arrangements, Ste12 binding on mating promoters is affected if the binding site carries point mutations (PRE-like) as is found in a majority of endogenous promoters which harbor a combination of a consensus PRE and a PRE-like site (p*AGA1*, p*FIG1*) [24]. A single base change to the TGAAAC consensus sequence can have a very different effect on the Ste12 affinity. While TaAAAC has only a 20% decrease in affinity, TGAgAC or TcAAAC result in a more than 95% reduction in competitive binding relative to the consensus sequence [27].

Therefore, to test the influence of the strength of the PRE site on the expression output, two PREs spaced by 3 base pairs were used and the sequence of one of the binding sites was modified. A clear decrease in the fraction of responding cells and in the level of expression are observed for the six PRE-like variants tested (Fig 3A and 3B). This is in line with previous measurements where the lowering of the affinity of the DNA-binding site results in a weaker gene expression output [27,30]. We can imagine that the lower affinity of the site decreases the residence time of Ste12 on the promoter and therefore limits the expression output. Importantly, however, the dynamics of gene expression is not influenced by the lowering of the binding site affinity and all the promoters tested here, despite their lower induction level, are induced rapidly (Fig 3D).

As hypothesized in a previous study, the association of Ste12 on promoters containing suboptimal PRE-arrangements seems to be enhanced by Kar4 [17]. Indeed, late pheromone responsive promoters like p*FIG1* show minimal expression in *kar4Δ* cells, while fast ones are independent of Kar4. Therefore, to evaluate the contribution of Kar4 to the expression output of synthetic promoters containing PRE sites with different affinities for Ste12, we measured a set of promoters in *kar4Δ* cells. On promoters containing a PRE-like site, our measurements suggest that Kar4 has a dual role. On the one hand, Kar4 contributes to the rapid activation of Ste12-bound promoters, while, on the other hand, it limits the level of induction of the promoter (Figs 3C, 3D, S6A and S6B). A promoter with two PRE spaced by 13 bp in tail-to-tail orientation follows the same trend of slow but high induction in *kar4Δ* cells, while the promoters with 5 and 15 bp in head-to-tail configuration are minimally affected by the same deletion (S6C and S6D Fig). Moreover, the high induction observed in *kar4Δ* cells for the promoter with a PRE-like seem to act via the activation domain of Ste12 because in the Ste12-EV chimeric protein, we don't see an effect of the deletion of *KAR4* (S6E and S6F Fig).

Taken together, these results indicate that the combination of a PRE with a PRE-like on a promoter tends to decrease the transcriptional output compared to two PRE sites, while the speed of gene induction remains unaffected, thanks to the contribution of Kar4. However, the phenotype associated with the deletion of KAR4 is complex because in the absence of this gene, the dynamics of induction of our synthetic promoters are slowed down, while their expression output is increased.

## Location of PRE sites on the promoter

In endogenous promoters, the distance of the PRE sites relative to the core promoter can vary greatly from -150 (p*FUS1*) to -440 p*STE12*. To test the influence of this parameter, we have moved the two PREs from p*SYN*$_{3TT}$ from their original position at -223 from the Start codon between -183 to -413 bp. Extending the distance between the Ste12 binding sites and the core promoter leads to a general decline of the expression output of the promoter (Fig 4A and 4B). A dip in expression output and fraction of responding cells is observed at -365 bp, which could be due to the presence of a loosely

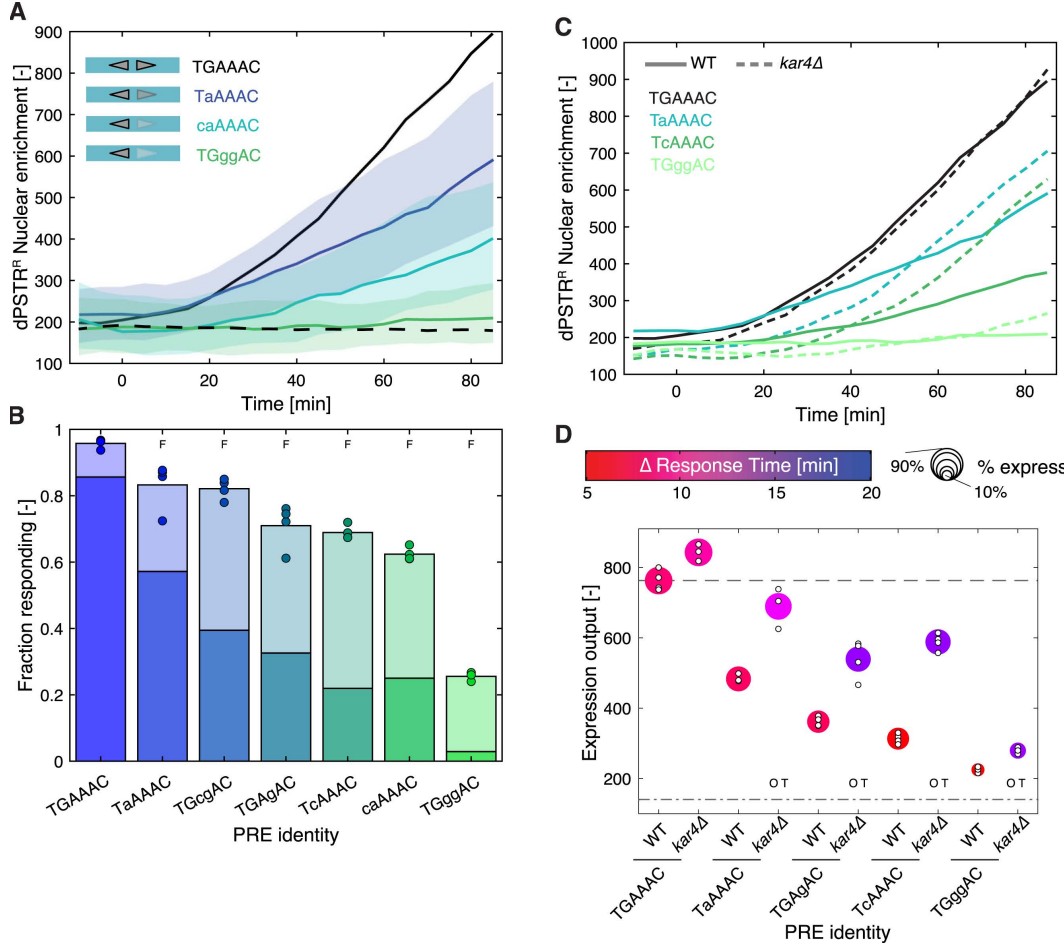

**Fig 3. Influence of Kar4 deletion on low affinity PRE sites. A.** Dynamics of nuclear relocation for the dPSTR$^R$ controlled by two PRE spaced by 3 bp in tail-to-tail orientation, where one of the PRE sequences was mutated to alter its binding affinity (colored lines: median of the population and shaded area: 25- to 75- percentile). The solid black line represents the median response of the consensus PRE (p$SYN_{3TT}$), while the dashed line represents the control promoter without PRE. **B.** Fraction of responding cells for the various PRE sequences. The darker portion of the bar represents the fraction of highly expressing cells (expression output > 50% of reference expression output) and the light bar the fraction of low-expressing cells (> 20% expression output < 50%). The marker represents the fraction of responding cells for individual replicates. The F indicates that the total fraction of expressing cells is significantly lower (t-test: p-val < 0.05) than the expression from the promoter with two consensus PRE sites. **C.** Dynamics of nuclear enrichment of WT (solid lines) and $kar4\Delta$ cells (dashed lines) with a synthetic promoter with 2 PREs spaced by 3 bp in tail-to-tail orientation and where one of the PRE sequences has been mutated to decrease the affinity to Ste12. **D.** Summary graph displaying the expression output, the speed and the fraction of responding cells for two PRE spaced by 3 bp in tail-to-tail orientation and where one of the PRE sequences has been mutated to decrease the affinity to Ste12 in WT and $kar4\Delta$ cells. The color of the marker indicates the difference in response time between the synthetic promoter and the reference p$AGA1$-dPSTR$^Y$. The size of the marker represents the fraction of responding cells. The expression output of individual replicates is indicated by small white dots. The dashed line represents the expression output and the dashed dotted line the expression threshold calculated based on the p$SYN_{3TT}$. The O and T indicate a significant difference between the mean of the replicates (t-test: p-val < 0.05) in the timing of induction (T) or in the expression output (O) between the WT and $kar4\Delta$ strains for the same promoter.

associated nucleosome. A similar decrease in expression has been observed for the Msn2 stress response TF where moving its binding sites away from the core promoter lowered the expression output of the promoter [42]. However, we demonstrate that this lower expression output is not correlated with a decrease in the speed of gene expression, since all the p$SYN$ tested remain fast (Fig 4C). Many of these promoters are in fact slightly faster than the reference p$SYN_{3TT}$, possibly because they have an increased basal expression level. In these synthetic constructs, the binding dynamics of

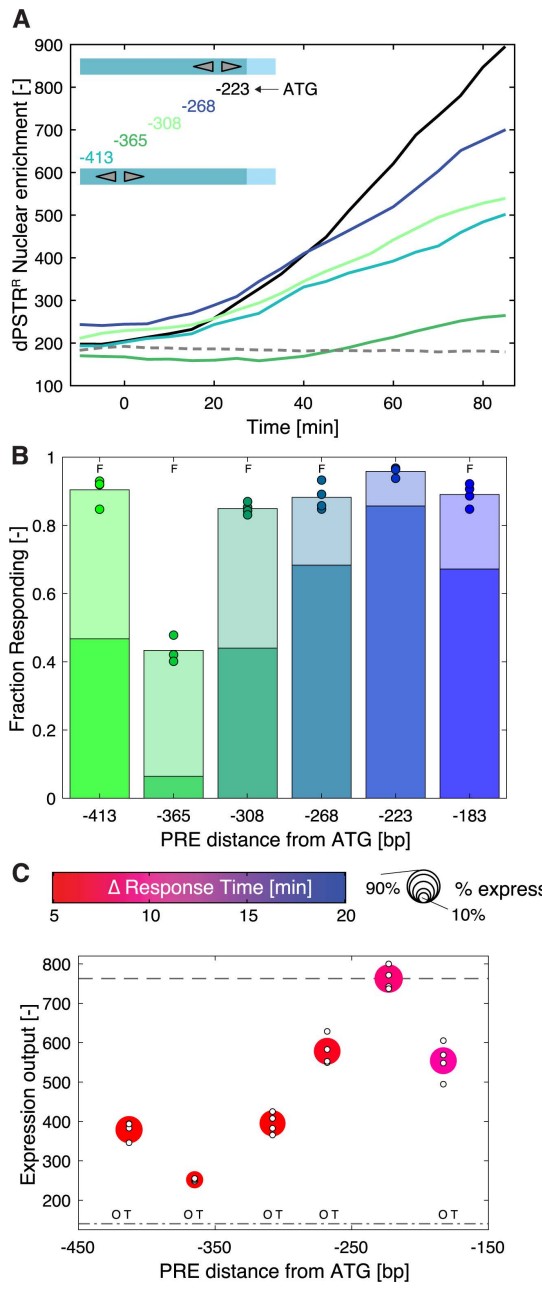

**Fig 4. Distance of the PREs to the core promoter impacts expression output in the synthetic promoter. A.** Dynamics of nuclear relocation for the dPSTR$^R$ controlled by two PREs spaced by 3 bp in tail-to-tail orientation placed at different positions on the promoter (color lines represent the median of the population). The solid black line represents the median response of the PREs placed 223 bp downstream of the start codon (p$SYN_{3TT}$), while the dashed line represents the control promoter without PRE. **B.** Fraction of responding cells for the various PRE sequences. The darker portion of the bar represents the fraction of highly expressing cells (expression output > 50% of reference expression output) and the light bar the fraction of low-expressing cells (> 20% expression output < 50%). The marker represents the fraction of responding cells for individual replicates. The F indicates that the total fraction of expressing cells is significantly lower (t-test: p-val < 0.05) than the expression from the reference construct with the two PRE placed at -223 bp. **C.** Summary graph displaying the expression output, the speed and the fraction of responding cells for PRE placed at various distances from the Start codon. The color of the marker indicates the difference in response time between the synthetic promoter and the reference p$AGA1$-dPSTR$^Y$. The size of the marker represents the fraction of responding cells. The expression output of individual replicates is indicated by small white dots. The dashed line represents the expression output and the dashed dotted line the expression threshold calculated based on the p$SYN_{3TT}$. The O and T indicate a significant difference between the mean of the replicates (t-test: p-val < 0.05) in the timing of induction (T) or in the expression output (O) relative to the p$SYN_{3TT}$.

Ste12 should not be modified by the location of the PREs on the promoter. Thus, we speculate that the increased distance between the Ste12 binding sites and the core promoter precludes the ability of Ste12 from activating the general transcription factors via the Mediator.

### Interplay between Ste12 and nucleosomes

The modulation of the distance between core promoter and PRE was also tested using a modified *AGA1* promoter where all the consensus binding sites were mutated, and a PRE-dimer was moved from -135 to -445 bp relative to the Start codon (Fig 5A). In this context, however, not only the expression levels but also the dynamics of induction were strongly influenced by the location of the PRE-sites (Fig 5B, 5C and 5D). We believe that this behavior can be explained by the position of the nucleosomes on the promoter. Indeed, the promoter activation is rapid and strong when the PREs fall in a nucleosome-depleted region (NDR). When the sites are located within a region protected by a nucleosome, the fraction of responding cells and the level and speed of induction are all attenuated. The most significant impact is observed with the PRE-dimers located at -330 bp, where Ste12 binding is conflicting with a nucleosome centered around -349 bp [43]. Interestingly, the fraction of expressing cells for this promoter is low (18%), but the few cells that express display a substantial nuclear enrichment of the dPSTR$^R$ (Fig 5E). This stochastic activation suggests the presence of a dynamic interplay between the binding of the nucleosomes and the Ste12 dimer. When the nucleosome is bound, no transcription takes place. In some cells, the histones can be displaced by the binding of Ste12, which remains stably associated to the promoter to induce a sizable but delayed expression.

To test this model more directly, the promoter sequence was mutated to introduce a nucleosome disfavoring sequence (poly dA/dT, 20 bp long) in the vicinity of the Ste12 binding sites (Fig 6A). The fraction of expressing cells increased from 20% to 50% when the dA/dT element was placed 6 bp away from the PRE sites (Fig 6B and 6C). An alternative option to increase the binding efficiency of Ste12 in the nucleosome bound region is to use multiple PRE sites. We used conformations with 3 PREs present in PRM1 and FUS1. The multiple sites and the high A/T content of the element facilitated the association of Ste12 to the promoter and resulted in an increased fraction of transcribing cells (Fig 6B and 6C). However, for all these constructs, the dynamics of induction were slow compared to the endogenous p*AGA1*. Interestingly, the addition of the dA/dT sequence slightly accelerated the induction of the reporter, suggesting that in a fraction of the population Ste12 can associate to the promoter under basal conditions (Fig 6D).

To probe the association of Ste12 on the DNA, Chromatin Immuno-Precipitation (Ch-IP) was performed on some of these promoter variants by amplifying a region centered around -259 bp. The basal association of Ste12 is much higher in the endogenous *AGA1* sequence than in our promoter variants (Fig 6E). In addition, we see a small but significant enrichment of Ste12 on the promoter when the dA/dT stretch is added 6 bp away from the two PREs compare located at -330. Upon pheromone treatment Ste12 becomes more than 10-fold enriched on the endogenous promoter while we don't detect a significant enrichment of the TF on the 2 PRE placed at -330 (Fig 6F). In contrast, promoters modified with the 3 PREs using the *FUS1* conformation or with the dA/dT stretch 6 bp away from the PREs, the enrichment of Ste12 is significant. In parallel, the association of histones in the same region of these promoters was evaluated by MNase protection assays. On the WT p*AGA1*, we observe a strong eviction of the nucleosomes 30 minutes after the stimulation of the cells with α-factor (Fig 6G). No significant eviction can be observed on the promoter with the 2 PRE at -330 bp, while the eviction is recovered when the dA/dT stretch is added 6 bp away from the two PREs or if three PREs are present. Both the recruitment of Ste12 and the eviction of the nucleosomes are in line with the higher expression output measured with the dPSTR$^R$. Note that under basal conditions, the nucleosome occupancy in this region is slightly lower for the promoter with 2 PRE at -330 bp, possibly because the position of the nucleosome is slightly shifted in this construct. These biochemical data confirm the hypothesis that the access to the PRE sites at -330 is limited for Ste12 under basal conditions which limits its ability to activate transcription from this site. However, the addition of a third PRE or the presence of a dA/dT stretch favors the access of Ste12 to the promoter by facilitating the eviction of the nucleosomes upon pheromone treatment.

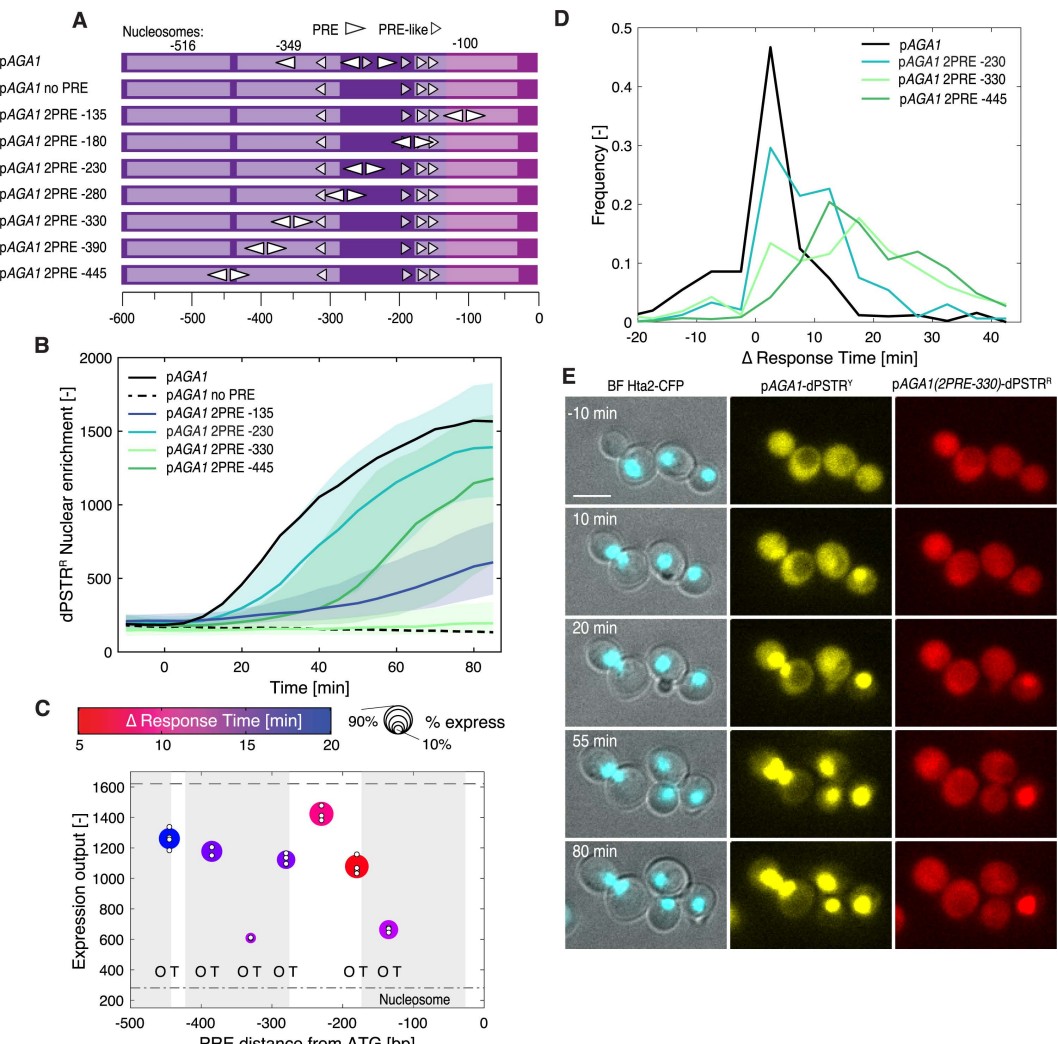

**Fig 5. Varying the location of the PRE dimer in an endogenous promoter can influence both level and dynamics of induction. A.** Scheme of the modified p*AGA1* promoters with consensus PRE sites indicated by elongated arrowheads and non-consensus PRE-like sites by small arrowheads. The regions protected by the nucleosomes positioned at -100, -349 and -516 are indicated by a lighter color. **B.** Dynamics of nuclear relocation for the dPSTR$^R$ controlled for a selected set of modified p*AGA1* promoters containing two PRE spaced by 3 bp in tail-to-tail orientation placed at different positions along the promoter (colored lines median of the population and shaded area 25- to 75- percentile). The solid black line represents the median response of the reference endogenous AGA1 promoter, while the dashed line represents a mutated p*AGA1* where the three PRE and one PRE-like sites are mutated. **C.** Summary graph displaying the expression output, the speed and the fraction of responding cells for the complete set of the modified p*AGA1* promoters. The color of the marker indicates the difference in response time between the synthetic promoter and the reference p*AGA1*-dPSTR$^Y$. The size of the marker represents the fraction of responding cells. The expression output of individual replicates is indicated by small white dots. The dashed line represents the expression output and the dashed dotted line the expression threshold calculated based on the p*AGA1*-dPSTR$^R$. The gray areas represent the regions of the promoters covered by the three nucleosomes. The O and T indicate a significant difference between the mean of the replicates (t-test: p-val < 0.05) in the timing of induction (T) or in the expression output (O) relative to the p*AGA1* with 2 PRE positioned at -230. **D.** Histograms of the difference in response time between a few selected modified p*AGA1*-dPSTR$^R$ and the reference p*AGA1*-dPSTR$^Y$ in the cells that express both constructs. **E.** Thumbnail images of cells bearing the p*AGA1*-dPSTR$^Y$ and the modified p*AGA1*-dPSTR$^R$ with the PRE dimer positioned 330 base pairs before the Start codon. While all cells in the image relocate the dPSTR$^Y$, only one cell displays a relocation of the dPSTR$^R$. The scale bar represents 5 μm.

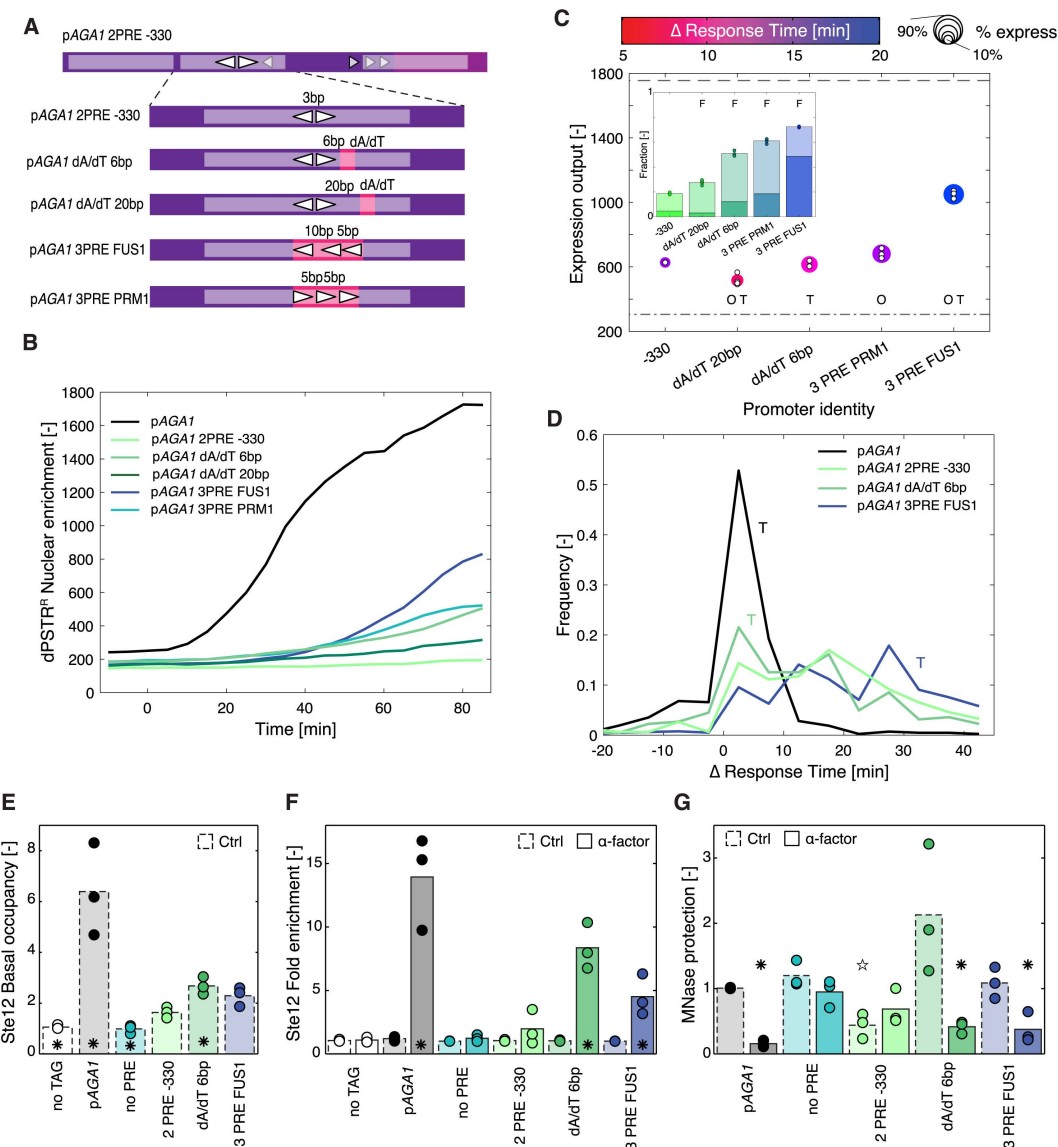

**Fig 6. Favoring nucleosome eviction increases expression output. A.** Scheme of the various promoters tested. Starting from the modified p*AGA1* with 2 PREs in tail-to-tail orientation at -330, poly-dA/dT segments of 20 bp were generated by mutations 6 or 20 bp away from the 2 PREs. Alternatively, the two PREs were replaced by three PRE sites extracted from the FUS1 or PRM1 promoters. **B.** Dynamics of nuclear relocation for the dPSTR^R controlled by modified p*AGA1* promoters. The four different modified promoters tested are shown in color. The solid black line represents the median response of the reference endogenous AGA1 promoter, while the dashed line represents the median of the population of cells with the 2 PREs in tail-to-tail orientation at -330. **C.** Summary graph displaying the expression output, the speed and the fraction of responding cells. The color of the marker indicates the difference in response time between the synthetic promoter and the reference p*AGA1*-dPSTR^Y. The size of the marker represents the fraction of responding cells. The dashed line represents the expression output and the dashed dotted line the expression threshold calculated based on the p*AGA1*-dPSTR^R. The O and T indicate a significant difference between the mean of the replicates (t-test: p-val<0.05) in the timing of induction (T) or in the expression output (O) relative to the p*AGA1* with 2 PRE positioned at -330. The inset represents the fraction of responding cells for each promoter. The dark portion of the bar represents the fraction of highly expressing cells (Expression output>50% EO of the reference promoter) and the light bar the fraction of low-expressing cells (20%<EO<50%). The marker represents the fraction of responding cells for individual replicates. The F indicates that the fraction of responding cells is significantly different relative to the p*AGA1* with 2 PRE positioned at -330. **D.** Histograms of the response time difference measured between the modified p*AGA1*-dPSTR^R and the reference p*AGA1*-dPSTR^Y. The solid black line corresponds to the endogenous AGA1 promoter, while the dashed line represents the 2 PREs in tail-to-tail orientation at -330. The addition of the poly-dA/dT 6 bp away from the PRE dimer (light green) slightly accelerates the induction of the promoter. Placing the three PREs of FUS1 (blue) slows down the induction of the promoter. The T indicates that the histograms for the two mutated promoters are significantly different from the endogenous promoter using a Wilcoxon rank sum test. **E.**

Basal association of Ste12 on various p*AGA1* promoters evaluated by Ch-IP in a region between -327 and -205. The values are normalized relative to the no TAG control. The star indicates a significant difference (t-test: p-val < 0.05) relative to the promoter with 2 PRE at position -330. **F.** Fold induction in Ste12 binding measured by Ch-IP after 30 min treatment with pheromone. The star indicates a significant difference between the untreated controls and the α-factor stimulated samples (t-test: p-val < 0.05). **G.** Eviction of nucleosomes monitored by MNase protection assays after 30min pheromone treatment in the same region as in E (between -327 and -205). Nucleosome occupancy is normalized relative to the untreated p*AGA1* endogenous promoter. A ∗ indicates a significant difference between the untreated control and the α-factor stimulated samples and the ☆ indicates a significant difference between the various mutated promoters in the untreated conditions relative to the p*AGA1* (t-test: p-val < 0.05).

The endogenous p*AGA1* promoter contains three consensus PREs and at least five PRE-like sites (S7A Fig). The PRE/PRE-like sites spaced by 3 bp at position -220 alone seem sufficient to ensure a high induction level. Therefore, it is not clear why there are so many additional Ste12 binding sites present in this sequence. One of their roles might be to increase the local concentration of Ste12 in the vicinity of the locus. Interestingly, if the first PRE at -196 bp or its neighboring PRE-like at -185 bp are mutated, the level of induction of the promoter remains almost identical, while the dynamics of induction are delayed (S7B and S7C Fig). Since these two PREs are 5 bp apart in a head-to-tail conformation, it is likely that they allow the formation of a Ste12 dimer. One possible explanation for the delayed expression observed is that these two binding sites contribute to defining the nucleosome depleted region in the p*AGA1* promoter which favors the basal association of Ste12 to the central PRE of the promoter. Regardless, together, these results demonstrate that nucleosomes prevent the basal association of Ste12 outside of the NDR. Sites present in a nucleosome protected region of the promoter will display slow induction kinetics and a low fraction of inducing cells. Additional PRE sites on endogenous promoters may contribute to shaping the NDR and thereby allowing indirectly an early activation of transcription.

## Discussion

The coding sequence of a protein can provide insights into many of its properties. However, predicting the level and the timing of its expression based on the promoter sequence remains a challenge [44]. Endogenous promoters display a large and complex palette of regulation combining binding sites for multiple TFs at various positions. Even in the simpler case of the mating pathway in budding yeast, where 200 genes are under the control of the single TF Ste12, the diversity in the organization of the PREs on these promoters is large. To elucidate the fundamental rules governing the induction of mating genes by Ste12, we designed synthetic promoters where the conformations of Ste12 binding sites could be tested systematically. To compare all these synthetic constructs, we placed them upstream of the core of the p*CYC1* promoter. Interestingly, a comparison between the p*AGA1* and the p*CYC1* core sequences indicate that this region also contributes to the regulation of the level and the speed of gene induction, which should be investigated further.

Based on these results we have identified various strategies that are at play in mating dependent promoters to regulate the properties of induction of a promoter. The level of induction can be controlled by two parameters: the binding affinity of the Ste12 sites and the distance from the core promoter. p*AGA1*, p*FAR1* and p*STE12* all possess 2 PRE spaced by 3 bp in tail-to-tail orientation. The PRE dimers is placed at -200 bp from the start in p*AGA1* while it is located at -300 bp and -400 bp for p*FAR1* and p*STE12* which could account for their lower inducibility [17]. The strength of Ste12 association on a promoter can be tuned by point mutation in the PREs or by changing the binding conformation. For instance, the two PREs spaced by 13 bp (-250 bp from ATG) in p*SST2* results in lower induction compared to p*AGA1*. p*KAR4* and p*AGA1* both have a PRE dimer spaced by 3 bp positioned ~200 bp from the start codon with one of the PRE which contains two point-mutations [24]. The stronger level of p*AGA1* induction suggests that the binding to the p*AGA1* PRE dimer is tighter. These comparisons clearly oversimplify the complexity of these promoters. All these endogenous sequences harbor numerous additional PRE or PRE-like sites and different core promoter sequences that together contribute to the final

inducibility of the promoter. However, our findings allow the identification of which PRE and/or PRE-like on a promoter are in the proper configuration to allow the formation of a Ste12 dimer.

### Role of Kar4

Previous measurements have suggested that Kar4 interacts directly with the DNA and that late mating genes contain specific binding sites for Kar4 [18]. However, we favor a model where Kar4 is recruited to the promoters via its interaction with Ste12. Multiple evidences point in this direction. We have shown previously that Kar4 is recruited to the promoters of both early and late genes [17]. The yeast epigenome project has identified a binding site for Kar4 on DNA which corresponds to the one of Ste12 [43]. Recent point mutants of Kar4 which display mating deficiencies have a decreased interaction with Ste12 [45].

Our present data demonstrate that Kar4 has a dual effect on some of our synthetic promoters. First, deleting *KAR4* results in a delayed induction for promoters with lower affinity Ste12 binding sites. This suggests that Kar4 contributes to the stabilization of Ste12 on these weaker sites under basal conditions such that upon stimulation of the pathway, these promoters can be activated promptly. In parallel, we observe an increase in the level of induction from these same promoters containing PRE-like sites in *kar4Δ* cells. This surprising behavior could act via the activation domain of Ste12 since we don't observe a similar increase in cells expressing the Ste12-EV chimeric TF in the absence of *KAR4*.

Absence of Kar4 slows down the induction of intermediate genes (*PRM1*, *FIG2*) and is required for the activation of late genes such as *FIG1* or *KAR3* [17]. These late promoters bear PRE-like combined with PRE which are hidden by nucleosomes. In the absence of Kar4, the binding of Ste12 to the promoter is presumably too weak to be stabilized and displaced by the nucleosomes without resulting in productive transcription whenever it binds.

### Dynamics of promoter induction

We identified two strategies to achieve a slow induction of the promoter: either by restricting the access of Ste12 to the promoter by nucleosomes or by using an unconventional PRE dimer (23 bp tail-to-tail or 15 bp head-to-tail or head-to-head). In the two dozen mating-dependent promoters that we have analyzed, we identified only one with 15 bp spacing. Thus, the nucleosome protection may be the preferred option, because it allows tuning both the speed and level of induction. Indeed, the large spacing between two PRE sites can only achieve a slow and low level of gene expression. In contrast, when the PRE dimer is moved within the p*AGA1* promoter delayed but high induction can be generated. Positioning the PRE dimer under the nucleosome slows down the induction but also strictly reduces the fraction of responding cells. However, placing the binding sites closer to the edge of a nucleosome protected region seems to be sufficient to obtain a similar decrease in the speed of induction without affecting severely the expression level and, importantly, the fraction of responding cells. In the promoter of the late gene *FIG1*, a PRE dimer (5 bp, tail-to-head) essential for gene expression is also placed at the boundary of the sequence protected by the nucleosome [17]. This positioning of the PRE dimers delays the induction of *FIG1* until it is required for the fusion of the two mating cells without compromising its robust expression when it is needed. Positioning the PREs further into the nucleosome protected region could lead to a stochastic activation of the protein which could be detrimental for the mating outcome.

The timing of gene expression is crucial for many processes, from the rapid induction of stress response genes to the controlled induction of proteins during the cell cycle. Cell-fate decisions are characterized with a chronology of gene expression which requires the ability to tune both the level and the dynamics of gene expression independently of each other. Regulating the basal association of Ste12 on promoters via the positioning of nucleosomes offers the opportunity to control the chronology of gene expression during the mating process. With our improved understanding of the regulation of the mating gene expression, we are in a better position to perturb the timing of key proteins implicated in mating to verify the importance of this chronology on the mating process.

## Materials and methods

### Yeast strains and plasmids

The synthetic promoters were derived from a plasmid containing a slightly modified p*AGA1*-dPSTR$^R$ plasmid [17] which contains a *ClaI* site at position 144 bp separating the regulatory region of *AGA1* (-1000 to -150) and the core region (-150–0). First the core promoter was replaced by the core promoter from *CYC1* (-180–0) cloned between *ApaI* and *ClaI*. Then the regulatory region (between *ClaI* and *AatII*) was replaced by fragments from a modified p*CYC1* promoter (138 bp) which destabilizes the association of nucleosomes [37]. This fragment encodes various conformations of Ste12 binding sites. The distance to the Start mentioned in the text and figures corresponds to the distance to the end of the *CYC1* or *AGA1* promoters. The actual Start of the inducible dPSTR moiety is 53 bp downstream due to the presence of multiple restriction sites. The longer spacing between Start and PRE sites (Fig 4) were obtained by duplicating the *CYC1* regulatory region. The core promoter and the multiple UAS sequences were synthesized as double stranded DNA fragments (IDT) or as plasmids (GeneScript). The list of synthetic sequences is provided in S1 Table. The plasmids obtained were verified by restriction digestion and sequencing and were transformed in the same reference strain from the W303 background with the histone Hta2 tagged with CFP (pGTH-CFP) [46] and containing the reference p*AGA1*-dPSTR$^Y$ [17]. The strains used in this study are listed in S1 Table.

Gene deletions were performed using the pFA6a KAN cassette [47]. Correct insertion of the deletion cassette was verified by PCR on genomic DNA. The human Estradiol receptor and the VP16 activation domain (EV) [39,40] were cloned in a pGT-NAT plasmid between *BamHI* and *NheI*. The Ste12-EV chimeric transcription factor was generated by transforming the EV sequence and NAT cassette with primers containing homology to the sequence flanking the activation domain of Ste12 (from 647 bp to STOP codon) in strains containing the selected synthetic promoters controlling the dPSTR$^R$. The transformants were selected on SD-UHL+NAT plates. The correct insertion of the EV sequence in frame with the Ste12 ORF was controlled by sequencing PCR fragments obtained from genomic DNA. The synthetic transcription factor Z4-EV is placed under the control of the RPL30 promoter [48]. The Venus is control by a modified p*GAL1* containing 6 Z4 binding sites. The pZ4$_{1BS}$ and pZ4$_{2BS}$ are based on the p*CYC1* promoter containing either one or two Z4 binding sites and driving the expression of the dPSTR$^R$.

Typically, eight transformants were selected and screened visually to verify the expression level of the three fluorescent constructs (Hta2-CFP, p*AGA1*-dPSTR$^Y$, p*SYN*-dPSTR$^R$). Four transformants were then quantified in time-lapse experiments and used as biological replicates for the measurements of one promoter. Based on the consistency of the p*AGA1*-dPSTR$^Y$ response and the behavior of the tested dPSTR$^R$, we excluded some of the replicates. If two or more replicates were not providing consistent results, the experiment was repeated with different transformants. In each experiment, the behavior of 500 cells is typically quantified, but some replicates contain only 200 cells and others up to 1000 individual cells. In the dPSTR nuclear relocation graphs and in the response time histograms, we represent the behavior of one representative replicate. In the bar graphs displaying the fraction of responding cells or in the summary graphs, the mean behavior of all the selected replicates is plotted.

### Time-lapse microscopy

Yeast strains were grown overnight to saturation in SD-full medium (Complete CSM DCS0031, ForMedium) diluted in the morning in fresh SD-full and grown for at least 4 hours. The cultures were diluted to OD 0.04 and briefly sonicated and 200 μl were loaded in the well of a 96-well plate (PS96B-G175, SwissCI) previously coated with Concanavalin A (L7647, Sigma-Aldrich). Cells settled in the well for 30 minutes before the start of the time-lapse.

Cells were imaged on an inverted wide-field epifluorescence microscope (Nikon, Ti2) enclosed in an incubation chamber set at 30° using a 40X Oil or a 40X AIR objective (Figs 2B to 2D, S3E, S3F, S5E, S5F and S6C to S6F). The fluorescence excitation is provided by a Lumencor Spectra 3 light source (LED intensity 50%). A CFP YFP RFP dichroic filter

PLOS Genetics

(F68-003) and appropriate emission filters were used to detect the fluorescence emission using a sCMOS camera (Hamamatsu, Fusion BT), with 50ms, 50ms and 100ms exposure time respectively. Up to 8 wells were imaged in parallel and 5 fields of view per well were monitored. Two brightfield images (one slightly out of focus) and the three fluorescent images were recorded every 5 minutes for 100 minutes. Before the third time point, 100μl of a 3μM α-factor solution in SD-full was added to each well to reach a final concentration of 1μM in the well. The α-factor is a gift from the Peter lab at the ETHZ. To induce the Ste12-EV construct, 100μl of a 3μM β-estradiol (Sigma-Aldrich, E2758-250MG) solution in SD-full was used to reach a final concentration of 1μM in the well.

## Data analysis

Time-lapse images were segmented and single cell data extracted using the YestQuant platform [49]. The CFP image allowed to determine the position of the nuclei of each cell. Using this first object, the two brightfield images were used to identify the border of each cell and define the cytoplasm object by removing the pixels belonging to the nucleus.

The quantification of the single cell traces was performed with Matlab (The Mathworks, R2023b). Only cells tracked from the beginning to the end of the timelapse were kept for the analysis. The nuclear enrichment is calculated as the difference between the average fluorescence of the nucleus object and the cytoplasm object. The basal nuclear enrichment is measured as the mean of the three first time points. The Expression Output (EO) corresponds to the difference between the maximum of a single cell traces and the basal level (S2D Fig).

To determine if a cell is deemed transcribing or not, two criteria are used. First, the last 5 points of the trace minus the basal level has to be significantly higher than zero (sign-test. P-value 0.05). Second, the Expression output of the trace (maximum of the trace – basal level) has to exceed a threshold. To define the threshold, we select one strain as the reference and calculate the threshold as 20% of the mean Expression ouput from all cells. The strain bearing the p*AGA1*-dPSTR$^R$ (Figs 5, 6, S2, S3 and S7) or the p*SYN*$_{3TT}$-dPSTR$^R$ (Figs 1, 3, 4 and S4) were used as references. Transcribing cells are further differentiated in strong and weak expression if their output exceed 50% of the Expression output of the reference strain and as weakly expressing if the expression output falls between 20% to 50% of the reference (S2D, S2E and S2F Fig).

The response time is defined as the first time point after the trace overcomes 20% of its own expression output (S2D Fig). The difference in response time between the tested dPSTR$^R$ and the internal reference construct p*AGA1*-dPSTR$^Y$ is only calculated for cells considered as transcribing both reporters (S2G and S2H Fig).

## ChIP assays

Yeast cultures were grown to early log phase (A$_{660}$ 0.4–0.6), then samples (50ml) were subjected to 1μM α-factor for 30 minutes. For crosslinking, yeast cells were treated with 1% formaldehyde for 20 minutes at room temperature. Glycine was added to a final concentration of 330mM for 15 minutes. Cells were collected, washed four times with cold TBS (20mM Tris-HCl, pH 7.5, 150mM NaCl), and kept at −20 °C for further processing. Cell pellets were resuspended in 0.3ml cold lysis buffer (50mM HEPES-KOH, pH 7.5, 140mM NaCl, 1mM EDTA, 0.1% sodium deoxycholate, 1% Triton-X 100, 1mM PMSF, 2mM benzamidine, 2 μg/mL leupeptin, 2 μg/mL pepstatin, 2 μg/mL aprotinin). An equal volume of glass beads was added, and cells were disrupted by vortexing (with Vortex Genie) for 13 minutes at 4°C. Glass beads were discarded and the crosslinked chromatin was sonicated with water bath sonicator (Bioruptor) to yield an average DNA fragment size of 350bp (range, 100–850bp). Finally, the samples were clarified by centrifugation at 16,000g for 5 minutes at 4°C. Supernatants were incubated with 50μL anti-HA 12CA5 monoclonal antibodies pre-coupled to pan mouse IgG Dynabeads (Invitrogen, 11042). After 120 minutes at 4°C on a rotator, beads were washed twice in 1mL lysis buffer, twice in 1mL lysis buffer with 500mM NaCl, twice in 1mL washing buffer (10mM Tris-HCl pH 8.0, 0.25 M LiCl, 1mM EDTA, 0.5% N-P40, 0.5% sodium deoxycholate) and once in 1mL TE (10mM Tris-HCl pH 8.0, 1mM EDTA). Immunoprecipitated

material was eluted twice from the beads by heating for 10 minutes at 65 °C in 50 µl elution buffer (25 mM Tris-HCl pH 7.5, 1 mM EDTA, 0.5% SDS). To reverse crosslinking, samples were adjusted to 0.3 ml with elution buffer and incubated overnight at 65°C. Proteins were digested by adding 0.5mg/ml Proteinase K (Novagen, 71049) for 1.5 hours at 37°C. DNA was extracted with phenol-chloroform-isoamyl alcohol (25:24:1) and chloroform. It was finally precipitated with 48% (v/v) of isopropanol and 90 mM NaCl for 2 hours at −20 °C in the presence of 20 µg glycogen, and resuspended in 30 µL of TE buffer. Quantitative PCR analysis on p$AGA1$-dPSTR-R used the following primers with locations indicated by the distance from the respective ATG initiation codon: $AGA1$ promoter (-310/-207); and $TEL$ (telomeric region on the right arm of chromosome VI). Experiments were done on three independent chromatin preparations and quantitative PCR analysis was done in real time using an Applied Biosystems Via7. Immunoprecipitation efficiency was calculated in triplicate by normalizing the amount of PCR product in the immunoprecipitated sample by that in $TEL$ sequence control. The binding data are presented as fold induction with respect to the non-treated condition, for basal binding of Ste12 the data are referenced to the untagged strain (no tag) which was set to 1.

### MNase nucleosome mapping

Yeast spheroplast preparation and micrococcal nuclease digestions were performed as described previously with modifications [50,51]. Ste12-6xHA tagged strain was grown to early log phase (A660 0.4–0.6) and samples of 500 ml of culture were exposed to 1 µM α-factor for 30 minutes. The cells were cross-linked with 1% formaldehyde for 15 minutes at 30°C and the reaction was stopped with 125 mM glycine for minutes. Cells were washed and resupended in 1M sorbitol TE buffer before cell wall digestion with 100 T zymoliase (USB). Cells were then lysed and immediately digested with 60–240 mU/µl of micrococcal nuclease (Worthington Biochemical Corporation, Lakewood; NJ., USA). DNA was subjected to electrophoresis in a 1.5% (w/v) agarose gel and the band corresponding to the mononucleosome was cut and purified using a QIAquick gel extraction kit (Qiagen). DNA was used in a real-time PCR with specific tiled oligonucleotides covering the $AGA1$ promoter of the dPSTR$^R$ (the endogenous AGA1 gene has been mutated). PCR quantification was referred to an internal loading control (telomeric region in chromosome 6) and nucleosome occupancy was normalized to 1 at the (-1) nucleosome region of the untreated condition. The degree of nucleosome eviction at the indicated region was set to 1 in the wild type strain in control conditions and used as a reference.

### Supporting information

**S1 Fig. Scheme of the mating pathway Yeast cells detect pheromone via a G-protein coupled receptor.** The G-protein disassemble and recruits the scaffold Ste5 at the plasma membrane. Then, Ste20 activates the MAP3K Ste11, which phosphorylates the MAP2K Ste7. Ste7 phosphorylates both MAPK Fus3 and Kss1. Fus3 phosphorylates Far1 to arrest the cell cycle in G1. Both Kss1 and Fus3 contribute to the transcriptional response by inhibiting the repression exerted on the TF Ste12 by Dig1 and Dig2.
(PDF)

**S2 Fig. Quantification of the expression response.** A. Dynamics of nuclear enrichment of the dPSTR$^R$ under the control of the p$AGA1$ (dark blue), the p$FIG1$ (magenta) or the synthetic promoter with two PREs in tail-to-tail orientation with 3 bp spacing (p$SYN_{3TT}$ - cyan). The solid line represents the median of the population, while the shaded area represents the 25- to 75-percentiles of the population. B. Dynamics of nuclear enrichment of the dPSTR$^Y$ under the control of the endogenous p$AGA1$ promoter which is present in parallel to the test dPSTR$^R$ reporter for the three strains presented in panel A. The response of the p$AGA1$-dPSTR$^Y$ serves as a control for the robustness of pheromone induction for all experiments. If the p$AGA1$-dPSTR$^Y$ is not induced properly, the experiment will be rejected. C. Correlation of the normalized expression level at 0, 20, 40, 60 min after the stimulus between the p$AGA1$-dPSTR$^Y$ (y-axis) and p$AGA1$ (left), p$SYN_{3TT}$ (middle) and the p$FIG1$ (right)-dPSTR$^R$ (x-axis). The Spearman correlation coefficient for each distribution is

indicated in the lower right corner. D. Description of the metrics measured from a single cell trace of nuclear enrichment. The mean of the nuclear enrichment of the first 3 time points is used to quantify the basal level of expression of the trace. The difference between the maximum of the trace and the basal level represents the expression output (EO). When the trace overcomes the threshold set by the 20% of this EO added to the basal level, the response time (RT) is defined. E. To characterize individual single cell traces as not responding, weakly or strongly responding, the mean EO of all the cells of a reference strain is used. In the present case, the reference strain is the p*AGA1*-dPSTR$^R$ construct which is used as a reference. Two criteria are used to define expressing cells. First, the last 5 points of the trace have to be significantly higher than the basal level (sign-test, blue, red, yellow traces). Second, the Expression Ouput of the trace has to overcome the expression threshold set at 20% of the reference trace EO (blue, red, green). The yellow and green traces which satisfy only one condition are thus considered as not expressing. In addition, if the expression output of a single cell exceeds 50% of the reference EO, it is considered as a strongly expressing cell (blue), while if it falls between the 20% to 50% it is defined as weakly expressing (red). F. Fraction of responding cells for the dPSTR$^R$ (left) and the dPSTR$^Y$ (right). The dark bar represents the mean fraction of strongly responding cells and the light bar the weakly expressing ones. The round markers represent the total fraction of responding cells measured in the 2–4 biological replicates used to build the graph. G. Scheme describing the calculation of the difference in response time between the p*AGA1*-dPSTR$^Y$ (top panels) and the test p*SYN*$_{3TT}$-dPSTR$^R$. In the left panels, three traces are shown with various delays in p*AGA1*-dPSTR$^Y$ induction and matching dynamics in the dPSTR$^R$ resulting in the calculation of a small difference in response time ($\Delta$RT). On the right panels, two traces in the p*AGA1*-dPSTR$^Y$ display a fast response while the dPSTR$^R$ responses rise much later, resulting in a large $\Delta$RT. H. Histogram of the difference in response time ($\Delta$RT) calculated for all the cells expressing both the reference p*AGA1*-dPSTR$^Y$ and the test dPSTR$^R$ controlled by p*AGA1* (blue), p*SYN*$_{3TT}$ (cyan) and p*FIG1* (magenta). (PDF)

**S3 Fig. Development of a synthetic mating-responsive promoter.** A. Scheme of the various promoter configurations tested starting from the p*AGA1* endogenous reporter and exchanging the core promoter and testing regulatory regions with a UAS containing various configurations of PREs. B. Dynamics of nuclear enrichment for the dPSTR$^R$ under the control of various synthetic promoters. The colored solid lines represent the median of the population and the shaded area, the 25- and 75-percentile of the population. The solid black line is the reference induction from the endogenous promoter p*AGA1* and the dashed line is the control promoter without PRE sites. C. Fraction of strongly (dark bar) and weakly (light bar) responding cells relative to the p*AGA1*-dPSTR$^R$. The total fraction of responding cells from individual replicates is displayed by the markers. D. Histogram of the difference in response time between the tested promoter and the internal reference provided by the p*AGA1*-dPSTR$^Y$. E. Dynamics of nuclear enrichment for the p*AGA1*-dPSTR$^Y$ (left) and p*SYN*-dPSTR$^R$ (right) variants in WT (solid line) and *ste12*$\Delta$ (dashed lines) strains. F. Fraction of responding cells in the WT and *ste12*$\Delta$ strains for the p*AGA1*-dPSTR$^Y$ (left) and p*SYN*-dPSTR$^R$ (right) variants. (PDF)

**S4 Fig. Influence of PRE orientation and spacing on the output of the synthetic promoter.** A, B and C. Time course of the nuclear enrichment of the dPSTR$^R$ for various distances of PRE placed in tail-to-head conformation towards the core (A), in tail-to-head conformation away from the core (B) and in head-to-head conformation (C). Three spacings are plotted in color. The solid lines represent the median and the shaded area the 25- to 75- percentile of the population. Gray lines represent the median of non-functional PRE conformations. The solid black line represents the median of the p*SYN*$_{3TT}$ reference promoter. The black dashed line is the median of the control synthetic promoter without PRE. D, E and F. Summary graph displaying the expression output, the speed and the fraction of responding cells for various spacings of the two PREs placed in tail-to-head conformation towards the core (D), in tail-to-head conformation away from the core (E) and in head-to-head conformation (F). The color of the marker indicates the difference in response time between the synthetic promoter and the reference p*AGA1*-dPSTR$^Y$. The size of the marker represents the fraction of responding cells.

The dashed line represents the expression output and the dashed dotted line the expression threshold calculated based on the p$SYN_{3TT}$. The O and T indicate a significant difference between the mean of the replicates (t-test: p-val < 0.05) in the timing of induction (T) or in the expression output (O) relative to the p$SYN_{3TT}$.
(PDF)

**S5 Fig. Influence of binding site number for the β-estradiol-dependent induction by Z4-EV or Ste12-EV.** A. Dynamics of nuclear enrichment of the dPSTR[R] under the control of synthetic promoters with one (blue) or two (green) Z4 binding sites (McIsaac *NAR* 2013) using the synthetic transcription factor Z4-EV upon stimulation with 1μm β-estradiol at time 0. B. Increase in cellular fluorescence as function of time for the reference promoter containing 6 Z4 binding sites and driving the expression of a Venus fluorescent protein which serves as an induction control for the experiment plotted in panel A. C. Dynamics of nuclear enrichment of the dPSTRR by the Ste12-EV upon stimulus by β-estradiol at time 0 under the control of different promoters containing zero (dark green), 1 PRE (light green) or two PREs in tail-to-tail orientation (blue). No detectable nuclear enrichment is observed for the 0 or 1 PRE controls, as well as, the 2 PRE spaced by 40 bp (light blue). If the 2 PREs are spaced by 3 bp (p$SYN_{3TT}$), the induction is strong (dark blue). D. Dynamics of nuclear enrichment of the control p*AGA1*-dPSTR[Y] by the Ste12-EV in the strains containing the synthetic promoters displayed in panel C.E. Comparison of the inducibility of p*AGA1*-dPSTR[Y] with the Ste12 WT (solid borders) or the Ste12-EV (dashed borders). The Expression Outputs for the p*AGA1* reporters induced by the Ste12 WT or the Ste12-EV were normalized relative to the Expression Output of the reference p$SYN_{3TT}$ sample. The bar represents the mean response of the replicates shown by the circles. A significant difference between the normalized EO Ste12-WT and Ste12-EV is indicated by a star (t-test: p-val < 0.05) F. Fraction of responding cells for the p*SYN*-dPSTR[R] variants with the Ste12 WT (solid borders) or the Ste12-EV (dashed borders). The bar represents the mean response of the replicates shown by the circles. A significant difference between the fraction of responding cells between Ste12-WT and Ste12-EV is indicated by a star (t-test: p-val < 0.05).
(PDF)

**S6 Fig. Effect of the deletion of KAR4 on mating gene induction.** A. Dynamics of nuclear enrichment of the p*AGA1*-dSPTR-Y in WT (solid lines) and *kar4Δ* cells (dashed lines). B. Histograms of the difference in response time between the tested promoter and the internal p*AGA1*-dPSTR[Y] reference for WT (solid lines) and *kar4Δ* cells (dashed lines) for two different non-consensus PRE sequences associated to one consensus PRE. C. Summary graph displaying the expression output, the speed and the fraction of responding cells for promoters with various PRE conformations in WT and *kar4Δ* cells. The color of the marker indicates the difference in response time between the synthetic promoter and the reference p*AGA1*-dPSTR[Y]. The size of the marker represents the fraction of responding cells. The expression output of individual replicates is indicated by small white dots. The dashed line represents the expression output and the dashed dotted line the expression threshold calculated based on the p$SYN_{3TT}$ in WT cells. The O and T indicate a significant difference between the mean of the replicates (t-test: p-val < 0.05) in the timing of induction (T) or in the expression output (O) between the WT and *kar4Δ* strains for the same promoter. D. Dynamics of nuclear enrichment of the p*SYN*-dPSTR[R] variants (right panel) and p*AGA1*-dSPTR-Y (left panel) in WT (solid lines) and *kar4Δ* cells (dashed lines). E. Dynamics of nuclear enrichment of the p*SYN*-dPSTR[R] with two PRE spaced by 3 bp in tail to tail orientation with one mutated PRE (TcAAAC) in WT (solid lines) and *kar4Δ* cells (dashed line) with the chimeric Ste12-EV promoter and stimulated with β-estradiol at time 0. F. Expression output of the strains measured in panel C. The O indicates that the p$SYN_{3TT}$ expresses significantly stronger than the two strain with the mutated PRE, which both express to the same level.
(PDF)

**S7 Fig. Mutation of the PRE1 and its associated PRE-like in p*AGA1* slows down response time.** A. Scheme of the p*AGA1* endogenous promoter, which contains three consensus PRE sites and at least five non-consensus ones.

PRE2 together with a non-consensus PRE spaced by 3 bp in tail-to-tail orientation are essential for the inducibility of the promoter. PRE1 (closest to the core) is spaced by 5 bp from a non-consensus site in tail to head conformation. PRE1 or its associated PRE-like have been mutated. B. Dynamics of nuclear enrichment of the dPSTR$^R$ under the control of the endogenous (dark blue) or the mutated (light blue or magenta) *AGA1* promoter. The solid line represents the median of the population and the shaded area the 25–75- percentile of the population. C. Histogram of the difference in response time between the tested promoters and the internal p*AGA1*-dPSTR$^Y$ reference. The T indicates that the histograms for the two mutated promoters are significantly different from the endogenous promoter using a Wilcoxon rank sum test.
(PDF)

**S1 Table. Strains Plasmids and Primers Supplementary File.**
(XLSX)

**S1 Data. Numerical values underlying the figures.**
(XLSX)

## Acknowledgments

We thank members of the Pelet lab for critical comments on the project. We thank Michael Taschner and Stephan Gruber for helpful discussions, Marta Schmitt, Stella Parzanese Yaima Matas and Mònica Romo for technical help

## Author contributions

**Conceptualization:** Sandrine Pinheiro, Serge Pelet.

**Data curation:** Mariona Nadal-Ribelles.

**Funding acquisition:** Francesc Posas, Serge Pelet.

**Investigation:** Sandrine Pinheiro, Mariona Nadal-Ribelles, Carme Solé, Vincent Vincenzetti, Yves Dusserre, Serge Pelet.

**Methodology:** Sandrine Pinheiro, Carme Solé.

**Resources:** Vincent Vincenzetti, Yves Dusserre.

**Supervision:** Francesc Posas.

**Writing – original draft:** Serge Pelet.

**Writing – review & editing:** Sandrine Pinheiro, Mariona Nadal-Ribelles, Francesc Posas.

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
