## [Decision Letter · Decision Letter 0]

Dear Dr Pelet,

Thank you very much for submitting your Research Article entitled 'Basal association of a transcription factor favors early gene expression' to PLOS Genetics.

The manuscript was fully evaluated at the editorial level and by two independent peer reviewers. The reviewers appreciated the attention to an important problem, but raised some substantial concerns about the current manuscript. Based on the reviews, we will not be able to accept this version of the manuscript, but we would be willing to review a much-revised version. We cannot, of course, promise publication at that time.

Both reviewers have several comments and suggestions. We would like you to address all of the reviewers' comments. In particular, for  Reviewer 1: 1) obtain mechanistic information (data) to explain how Kar4 might be important for quick transcriptional activation by Ste12, and 2) the request of performing CHIP assays (or equivalent) to determine Ste12 binding relative to nucleosome positioning. Regarding Reviewer 2, please address all comments, with particular care to reassess the conclusions/interpretations derived of the data presented.

If you decide to revise the manuscript for further consideration at PLOS Genetics, please aim to resubmit within the next 60 days, unless it will take extra time to address the concerns of the reviewers, in which case we would appreciate an expected resubmission date by email to plosgenetics@plos.org.

If present, accompanying reviewer attachments are included with this email; please notify the journal office if any appear to be missing. They will also be available for download from the link below. You can use this link to log into the system when you are ready to submit a revised version, having first consulted our Submission Checklist .

PLOS has incorporated Similarity Check , powered by iThenticate, into its journal-wide submission system in order to screen submitted content for originality before publication. Each PLOS journal undertakes screening on a proportion of submitted articles. You will be contacted if needed following the screening process.

To resubmit, log into your Editorial Manager account and select the option 'Revise Submission' in the 'Submissions Needing Revision' folder.

We are sorry that we cannot be more positive about your manuscript at this stage. Please do not hesitate to contact us if you have any concerns or questions.

Yours sincerely,

Monica Colaiácovo

Section Editor

PLOS Genetics

Alejandro Colman-Lerner

Guest Editor

PLOS Genetics

Reviewer's Responses to Questions

**Comments to the Authors:**

Reviewer #1: In this manuscript, the authors examine kinetic response of a series of artificial reporter genes regulated by the mating pheromone responsive transcription factor Ste12 in yeast. For these studies they used a dynamic Protein Synthesis Translocation Reporter (dPSTR) which enables reliable measurement of gene expression using fluorescent reporter proteins. The veracity of this system was initially verified using the pheromone responsive AGA1 promoter. They then examine response of a variety of synthetic promoter constructs using the CYC1 core promoter element where they examine variations in organization of pheromone response elements (PREs) at upstream locations. From their studies they conclude that initial rapid induction of pheromone responsive promoters is encouraged by basal association of a Ste12 dimer in a nucleosome depleted upstream promoter region.

Overall, this is an interesting study which examines the effect of Ste12 binding site positioning on kinetics of gene expression in response to pheromone. Ste12 is regulated by MAPK signaling in yeast, and represents an important model transcription factor for growth factor regulated gene expression in eukaryotes. The system designed to examine effects of PRE spacing and orientation for kinetic analysis of expression is quite elegant and the results are mostly clearly described. A similar study detailing effects of PRE positioning for pheromone responsive expression of artificial promoters was previously described, but only using single time point measurements (Ref. 25), This report extends upon these observations using more detailed kinetic analysis. However, a limitation of this study is that the central conclusions are speculative in that the authors have not conclusively demonstrated that rapid induction of pheromone responsive promoters are mediated by a Ste12 protein dimer or within a nucleosome depleted region. Furthermore, I am not sure how relevant the results are for explaining overall pheromone response, considering that Ste12 interacts with multiple additional DNA binding factors, including at least MCM1, a1, alpha2, and likely others. This is not discussed anywhere in the manuscript. Also, all of the conclusions are based on reporter gene expression assays, and there are no results presented to quantify interaction of Ste12 with the artificial promoters, or verify nucleosome occupancy. Consequently the major conclusions are based on vastly over interpreted observations.

Specific comments:

- The effect of PRE conformation for induction of pheromone responsive genes is complicated and could even be affected by which core promoter element is used, this is not examined in the present study (this is just a comment, not a suggestion for additional data). As mentioned above, response to pheromone involves interaction of Ste12 with multiple additional proteins, some of which have likely not been identified. This should be clarified in the text.

- Relating to the above comment, in Figure 2A, the authors show schematics of Ste12 protein bound to 2 PREs. Are these illustrations based on structural analysis or simply speculation? This should be clarified.

- The results examining the effect of Kar4 are confusing. First of all, Kar4 is thought to bind to its own cis-element on a subset of pheromone response genes, this is not discussed in the text. Furthermore, the observation that Kar4 contributes to rapid activation of the artificial promoters is contrary to its known role for induction of delayed genes during pheromone response. Without some mechanistic information regarding interaction of Ste12 and Kar4 protein, the significance of these results are questionable.

- The conclusions are based on the presumed positioning of nucleosomes and binding of Ste12, this would be strengthened considerably with biochemical analysis using ChIP-PCR, for any of the experiments described, including those involving insertion of the putative nucleosome displacing sequences in the synthetic promoter. This would be important to support statements as follows "when the nucleosome is bound, no transcription takes place. In some cells, the histones can be displaced by the binding of Ste12, which remains stably associated to the promoter to induce a sizable but delayed expression".

Minor comments:

- The authors need to use proper genetic annotation throughout the manuscript, genes should be noted in italics.

- Page 8, the phrase "slow mating promoters", should probably read something like delayed pheromone responsive promoters.

Reviewer #2: This work analyzes how promoter architecture dictates the pattern of transcriptional induction. The system being studied is the transcriptional response to yeast mating pheromone, which involves a transcription factor Ste12 and its DNA binding sites known as PREs (pheromone response elements). To help shed light on what determines why different transcripts are induced with different kinetics, magnitudes, and/or population heterogeneities, the authors take a systematic approach involving the construction of synthetic reporter constructs. These synthetic reporters allow the authors to dissect the key determinants by varying individual promoter features, including the number of PREs as well as their spacing, orientation, sequence optimization, distance to transcriptional start site, and positioning relative to nucleosome-occupied regions. The results convincingly show that each of these parameters makes important contributions to the transcriptional response behavior.

Overall, the experiments are sound and the data are of high quality. A primary strength of the study is that the systematic approach allows the individual features of promoter architecture to be isolated from the others, which provides confidence that the change in expression pattern can be properly attributed to the feature being varied. The main weakness of the manuscript is that there are several places where the authors do not appropriately distinguish between observations and interpretations; that is, they make statements about parameters that they did not actually measure, and hence the claims do not seem justified. This leads to some reservations about whether the findings are overinterpreted.

Below are a variety of issues that I think would be valuable for the authors to elaborate upon, clarify, or reinterpret.

1. There are several places where findings and interpretation should be disentangled:

(a.) On page 8, a Section Heading states "Kar4 stabilizes Ste12 binding to the promoter". In fact, there are no measurements of Ste12 binding, and no measurements of Kar4 stabilization of this binding. The text should more accurately state and emphasize what is demonstrated, and then describe how the findings might be interpreted.

(b.) In the same section on pg 8 and the relevant Figure 3, there are six PRE variants with deviations from the consensus sequence. These are described as having different Ste12 binding affinities (e.g., Fig 3 legend title: "low affinity PRE sites"). But the authors do not explicitly state if the Ste12 binding affinities for these specific sequences has been previously measured, and, if so, whether the rank order of binding strength fits with the progressive changes in transcriptional magnitude observed. Consequently, it is unclear if the interpretation of these experiments is based primarily on supposition or on established evidence for changes in affinity.

(c.) Pg 11, top: "...these results demonstrate that nucleosomes prevent the basal association of Ste12 outside of the NDR". The claim of what the results "demonstrate" seems like an overinterpretation. This study contains no measurements of basal association or nucleosomal occupancy. The text should more accurately describe what was measured and observed, and then describe the basis for the eventual interpretation (and the basis for confidence in its likelihood). Even if the interpretation is reasonable, it seems inappropriate to muddy the distinction.

2. Page 7, middle: "Additionally, Ste12 controls its own expression, thus the level of the chimeric transcription factor is probably lower than the ones of the endogenous protein in absence of beta-estradiol." The logic here is not compelling. Just because Ste12 increases its own expression, there is no inherent basis for expecting that the eventual levels are greater rather than lower than the beta-estradiol induced levels. For example native STE12 transcripts might increase from 0.1x to 0.2x of the b-estradiol-induced levels, or from 0.5x to 1.0x, or 1.0x to 2.0x, or 5x to 10x. So, the authors' claim that it is "probably lower" does not seem justified; it could be lower, higher, or equal.

3. Figure 5C needs better explanation in the legend. It shows 7 data circles, presumably all from PRE-containing constructs. But in Fig 5A there are only 5 PRE-containing arrangements shown. What are the other 2 data circles in Fig 5C? The 2 that don’t seem represented in Fig 5A are positioned at roughly -275 and -380. This needs clarification.

4. Page 10 and Supp Figs 7B-C: The differences in induction dynamics between pAGA1, PRE1∆, and PRE-like∆ look very minimal and overlapping. On what basis should we interpret these as different, rather than as essentially indistinguishable? Was a statistical test applied?

Other minor points:

5. The numbers in Fig 2E are somewhat puzzling. Fig 2D suggests that the max expression from the 23bp-TT configuration is ~ 250, compared to ~700 for the 3bp-TT configuration. This would yield ~0.35, rather than ~0.5 (in Fig 2E). Can the authors clarify the possible source of the discrepancy? Also, specifiy the induction time point used to obtain the values in Fig 2E.

6. Is there a reason that the -135 construct in Figs 5A-B are not also represented in Fig 5D?

7. Page 9: "In this synthetic construct, where the nucleosome association has been inhibited...". This description is unnecessarily mysterious. It would be preferable to provide a brief description of HOW nucleosome association has been inhibited in this construct, especially because the putative nucleosomal occupancy will play an important role in later experiments.

8. Page 10, line 2: I suspect this should say "-330bp" (not "-320bp").

9. Page 5: It would be helpful to readers if acronyms such as "pSYN_3TT" are briefly defined, to help follow subsequent parts of the text. Presumably “3TT” here means 3bp separation, tail-to-tail. If so, state that directly.

**Have all data underlying the figures and results presented in the manuscript been provided?**

Reviewer #1: Yes

Reviewer #2: None

PLOS authors have the option to publish the peer review history of their article (what does this mean? ). If published, this will include your full peer review and any attached files.

**Do you want your identity to be public for this peer review?** For information about this choice, including consent withdrawal, please see our Privacy Policy .

Reviewer #1: **Yes: ** Ivan Sadowski

Reviewer #2: No

---

## [Decision Letter · Decision Letter 1]

PGENETICS-D-24-00568R1

Basal association of a transcription factor favors early gene expression

PLOS Genetics

Dear Dr. Pelet,

Thank you for submitting your manuscript to PLOS Genetics. After careful consideration, we feel that it has merit but does not fully meet PLOS Genetics's publication criteria as it currently stands. Therefore, we invite you to submit a revised version of the manuscript that addresses the points raised during the review process.

Please submit your revised manuscript within 30 days Apr 26 2025 11:59PM. If you will need more time than this to complete your revisions, please reply to this message or contact the journal office at plosgenetics@plos.org. Please include the following items when submitting your revised manuscript:

We look forward to receiving your revised manuscript.

Kind regards,

Alejandro Colman-Lerner, Ph.D.

Guest Editor

PLOS Genetics

Monica Colaiácovo

Section Editor

PLOS Genetics

Aimée Dudley

Editor-in-Chief

PLOS Genetics

Anne Goriely

Editor-in-Chief

PLOS Genetics

**Additional Editor Comments :**

Dear Dr Pelet,

As you can see, both reviewers are happy with your revised version, and I share that opinion as well. Thus, I think the paper should be accepted. However, before I can make that decision, I have a few comments of my own that I ask you to address.

1- Figure 2E. Your motivating question was: is the AD of Ste12 responsible (due to its ability to dimerize/multimerize) for the ability of Ste12 to induce transcription using PREs that are "far away" (like 15bp)? In the data, the chimeras with Ste12-DBD-ERE-VP16 show only a significant (but minor) reduction in output relative to WT Ste12 for the 15 and 23 bp separation. I would conclude that the answer to the motivating question is NO, since there is plenty of output even in the absence of Ste12AD. However, in the paper it seems that it is a YES.

Please comment/clarify, etc.

2- For Figure 6G (the new eviction experiment). It is not stated which primers were used in the qPCR to measure protection. Do they cover all the promoter or just the region where the PRE was placed? )(naively I thought it should be the region where the PREs were placed) Please include this information.

This is relevant, since the wt AGA1 promoter has the PREs in a region supposedly already mostly devoid of nucleosomes. But addition of alpha factor seems to deplete of nucleosomes anyway.

Also, the way it is normalized (to the value of pAGA1 without alpha factor), suggests that we should be able to compare protection in one promoter with the others? Why is there more protection in wt pAGA1 without pheromone than in 2 PRE -330?

Add the statistics to compare between promoters.

Minor comments:

1- The Zenodo link does not seem to be operational.

2- There seems to be missing statistical analysis on several Figures. For example, in all the transcription dynamics plots, such as 1D, 2C, 2D, there is no statistics to enable the reader to decide if one construct is faster than the other. Also on some of the output plots as well: no statistics on Fig3B, 4B.

3- In many cases, the number of biological replicates done for each experiments is not stated. Figure 3A is just as one example. In others the reader can count the number of dots, but it would be cleaner if stated in all legends.

Best

**Journal Requirements:**

1) Thank you for stating "Raw single cell traces are available on Zenodo: 10.5281/zenodo.14438716." We couldn't access the dataset. Please provide a new link to access the dataset or provide further details to locate the data.

2) Please ensure that the funders and grant numbers match between the Financial Disclosure field and the Funding Information tab in your submission form. Note that the funders must be provided in the same order in both places as well. Currently, "CERCA Programme of the Government of Catalonia" is missing from the Funding Information tab.

**Reviewers' comments:**

Reviewer's Responses to Questions

Reviewer #1: The authors have done a good job in addressing all of my comments from the previous version.

Reviewer #2: In this revised manuscript, the authors have satisfactorily addressed the issues raised in my original review. Moreover, the new data on Ste12 occupancy and nucleosome eviction in Figures 6E-6G are a very welcome addition; the results are convincing and they give substantial support to the overall story by showing that gene induction magnitude and timing are affected by features controlling the ability of Ste12 binding to compete with nucleosomal occupancy.

Overall, I found this to be an interesting and well-executed study. I have no serious concerns. There are a few minor points listed below that warrant correction or clarification.

1. Regarding Fig 4A-B, the text on page 10 states that "Extending the distance between the Ste12 binding sites and the core promoter leads to a general decline of the expression output...". Curiously, however, strong expression returns for the greatest disance (at -413), but the authors never mention this. Why is this not mentioned? And, what is the possible interpretation?

2. Regarding Fig 5A-C, the numbers for the PRE positions don't match between text and Figure. The text (pg 11) says "from -128 to -435", but Fig 5A-C says "-135" and "-445". It seems these should agree with each other, no?

3. I think Supplementary Fig 6C should have a key to explain the symbols (of the same sort is found in Figs 3D and 4C).

4. Minor typo: Pg 4, 9th line from bottom: "expression of" should say "expression or".

**Have all data underlying the figures and results presented in the manuscript been provided?**

Reviewer #1: Yes

Reviewer #2: None

PLOS authors have the option to publish the peer review history of their article (what does this mean? ). If published, this will include your full peer review and any attached files.

**Do you want your identity to be public for this peer review?** For information about this choice, including consent withdrawal, please see our Privacy Policy .

Reviewer #1: No

Reviewer #2: No

**Figure resubmission:**
---

## [Editor Report · Decision Letter 2]

Dear Dr Pelet,

We are pleased to inform you that your manuscript entitled "Basal association of a transcription factor favors early gene expression" has been editorially accepted for publication in PLOS Genetics. Congratulations!

Yours sincerely,

Alejandro Colman-Lerner, Ph.D.

Guest Editor

PLOS Genetics

Monica Colaiácovo

Section Editor

PLOS Genetics

Aimée Dudley

Editor-in-Chief

PLOS Genetics

Anne Goriely

Editor-in-Chief

PLOS Genetics

Comments from the reviewers (if applicable):

**Data Deposition**

http://datadryad.org/submit?journalID=pgenetics&manu=PGENETICS-D-24-00568R2

**Press Queries**

---

## [Editor Report · Acceptance letter]

PGENETICS-D-24-00568R2

Basal association of a transcription factor favors early gene expression

Dear Dr Pelet,

We are pleased to inform you that your manuscript entitled "Basal association of a transcription factor favors early gene expression" has been formally accepted for publication in PLOS Genetics! Your manuscript is now with our production department and you will be notified of the publication date in due course.

With kind regards,

Anita Estes

PLOS Genetics

On behalf of:
